# Comprehensive identification of mRNA isoforms reveals the diversity of neural cell-surface molecules with roles in retinal development and disease

Thomas A. Ray[1,2], Kelly Cochran[1,2], Chris Kozlowski [1,2], Jingjing Wang [1,2], Graham Alexander[3], Martha A. Cady, William J. Spencer, Philip A. Ruzycki[4], Brian S. Clark [4,5], Annelies Laeremans[6], Ming-Xiao He[6], Xiaoming Wang[6], Emily Park[6], Ying Hao[2], Alessandro Iannaccone[2], Gary Hu[1,2], Olivier Fedrigo [3,7], Nikolai P. Skiba[2], Vadim Y. Arshavsky [2] & Jeremy N. Kay [1,2 ✉]

Genes encoding cell-surface proteins control nervous system development and are implicated in neurological disorders. These genes produce alternative mRNA isoforms which remain poorly characterized, impeding understanding of how disease-associated mutations cause pathology. Here we introduce a strategy to define complete portfolios of full-length isoforms encoded by individual genes. Applying this approach to neural cell-surface molecules, we identify thousands of unannotated isoforms expressed in retina and brain. By mass spectrometry we confirm expression of newly-discovered proteins on the cell surface in vivo. Remarkably, we discover that the major isoform of a retinal degeneration gene, *CRB1*, was previously overlooked. This *CRB1* isoform is the only one expressed by photoreceptors, the affected cells in CRB1 disease. Using mouse mutants, we identify a function for this isoform at photoreceptor-glial junctions and demonstrate that loss of this isoform accelerates photoreceptor death. Therefore, our isoform identification strategy enables discovery of new gene functions relevant to disease.

[1] Department of Neurobiology, Duke University School of Medicine, Durham, NC 27710, USA. [2] Department of Ophthalmology, Duke University School of Medicine, Durham, NC 27710, USA. [3] Center for Genomic and Computational Biology, Duke University, Durham, NC 27710, USA. [4] John F. Hardesty, M.D. Department of Ophthalmology and Visual Sciences, Washington University, St. Louis, MO 63110, USA. [5] Department of Developmental Biology, Washington University, St. Louis, MO 63110, USA. [6] Advanced Cell Diagnostics, Newark, CA 94560, USA. [7] Present address: The Rockefeller University, 1230 York Avenue, New York, NY 10065, USA. ✉email: jeremy.kay@duke.edu

Most genes generate multiple mRNA isoforms. Mechanisms such as alternative splicing, intron retention, and alternative transcription start/stop sites serve to diversify mRNA sequences, yielding isoforms that often differ in their protein-coding capacity[1–4]. These mechanisms are especially common in the central nervous system (CNS), where alternative isoform use is particularly prevalent[1,5]. The diverse portfolio of CNS isoforms contributes in important ways to a wide range of neural functions[6–8]. Moreover, dysregulation of isoform expression is implicated in neurological disorders[9–11]. For these reasons, there is increasing awareness that genetic studies of CNS development, function, and disease will need to take isoform diversity into account.

Despite this clear importance, information about the number and the identity of CNS mRNA isoforms remains surprisingly scarce—even within the major transcriptome annotation databases[12]. RNA-sequencing (RNA-seq) has led to an explosion of new information about alternative splicing; however, because typical RNA-seq read lengths are <200 bp, the method is not able to resolve the full-length sequence of multi-kilobase transcripts. Therefore, by relying on RNA-seq alone, it is impossible to determine the number of isoforms produced by any given gene, or their full-length sequences.

In the absence of reliable full-length transcript annotations, the design and interpretation of genetic experiments become exceedingly difficult. For example, unless transcript sequences are known, it is difficult to be certain that a "knockout" mouse allele has been properly designed to fully eliminate expression of all isoforms. Unannotated isoforms can also be problematic for understanding how mutations lead to pathology in human genetic disease. Hidden isoforms may possess uncharacterized protein-coding sequences or unexpected expression patterns, which could cause the molecular and cellular consequences of disease-linked mutations to be misinterpreted. Thus, lack of comprehensive isoform sequence information remains a major impediment to our understanding of both normal gene function and the phenotypic consequences of gene dysfunction[12].

Here we sought to address this deficiency by uncovering the isoform diversity of genes encoding CNS cell-surface molecules. We focused on this gene set for two reasons. First, some of the most striking examples of functionally significant isoform diversity are found among genes of this class. These include the Drosophila Dscam1 gene[13], the mammalian clustered protocadherins[14], and the neurexin gene family[15,16]. Each of these genes produces hundreds of protein isoforms with distinct binding specificity, diversifying the molecular recognition events that mediate assembly of the nervous system[17–19]. From these examples it seems clear that, to understand the molecular basis for neural circuit wiring, it will be necessary to define the precise repertoire of cell-surface protein isoforms expressed in the developing CNS. A second reason for focusing on cell-surface molecules is that genetic alterations affecting them have been implicated in numerous CNS disorders. These include autism[20], epilepsy[21,22], and neurodegeneration[23–26]. However, in the vast majority of these cases, it remains unclear why certain mutations increase disease risk. Comprehensive isoform identification has great potential to reveal how these genetic variants cause disease pathology.

Here we devised a strategy that leverages Pacific Biosciences (PacBio) long-read sequencing technology to generate comprehensive catalogs of CNS cell-surface molecules. Long-read sequencing is ideal for full-length transcript identification; however, sequencing depth is not yet sufficient to reveal the full scope of isoform diversity[27–30]. To overcome this limitation we adapted a strategy from short-read sequencing, in which targeted cDNAs are pulled down with biotinylated probes against

known exons[31,32]. This approach yielded major improvements in long-read coverage, revealing an unexpectedly rich diversity of isoforms encoded by the targeted genes. To make sense of these complex datasets, we developed bioinformatics tools for the classification and comparison of isoforms, and for determining their expression patterns using short-read RNA-seq data.

To demonstrate how our approach can illuminate gene function, we analyzed one gene, Crb1, in detail. Crb1 is a member of the evolutionarily conserved Crumbs gene family, which encode cell-surface proteins that mediate apico-basal epithelial polarity[33]. In the retina, CRB1 localizes to the outer limiting membrane (OLM), a set of structurally important junctions between photoreceptors and neighboring glial cells known as Müller glia[26]. OLM junctions form at precise subcellular domains within each cell type, suggesting a high degree of molecular specificity in the establishment of these intercellular contacts[34]. There is great interest in understanding the function of CRB1 at OLM junctions, because loss-of-function mutations in human CRB1 cause a spectrum of retinal degenerative disorders[35]. It has been proposed that loss of OLM integrity might play a role in disease pathogenesis[26,36], but studies in mice have yet to convincingly support this model: Deletion of the known Crb1 isoform neither disrupts the OLM nor causes significant photoreceptor degeneration[37]. Here we identify a new Crb1 isoform that is far more abundant—in both mouse and human retina—than the canonical isoform. Using mutant mice, we show that this isoform is required for OLM integrity and that its removal is required to adequately phenocopy the human degenerative disease. These results call for a major revision to prevailing models of CRB1 disease genetics and pathobiology. Thus, our findings provide a striking example of how comprehensive isoform characterization can unveil important gene functions that were previously overlooked, enabling new insights into many biological questions including the biology of disease-associated genes.

## Results

**Cataloging isoforms via long-read capture sequencing.** To define the isoform diversity of CNS cell surface molecules, we first manually screened RNA-seq data from mouse retina and brain[38,39] to identify genes that showed unannotated mRNA diversity. We focused on cell surface receptors of the epidermal growth factor (EGF), Immunoglobulin (Ig), and adhesion G-protein coupled receptor superfamilies, as these genes have known roles in cell-cell recognition. For each gene screened ($n = 402$), we assessed whether it was expressed during CNS development, and if so, whether RNA-seq reads supported existence of unannotated exons or splice junctions (Fig. 1a). We found that ~15% of genes (60/402) showed strong evidence of multiple unannotated features. These genes were selected as candidates for long-read sequencing.

To comprehensively identify these genes' transcripts, we developed a method to improve PacBio sequencing depth for large (>4 kb) and moderately expressed cDNAs, such as those on our candidate gene list. We term this strategy long-read capture sequencing (lrCaptureSeq), because we adapted prior CaptureSeq approaches[31,32,40] to enable characterization of protein-coding cDNAs with the long-read PacBio platform. In lrCaptureSeq (Fig. 1b, c), biotinylated probes are designed to tile known exons without crossing splice junctions, so as to avoid targeting particular isoforms. These probes are used to pull down cDNAs from libraries that have been size-selected to filter truncated cDNAs. This size selection was essential to obtaining full-length reads (Supplementary Fig. 1a), because shorter fragments tend to dominate the sequencing output[15].

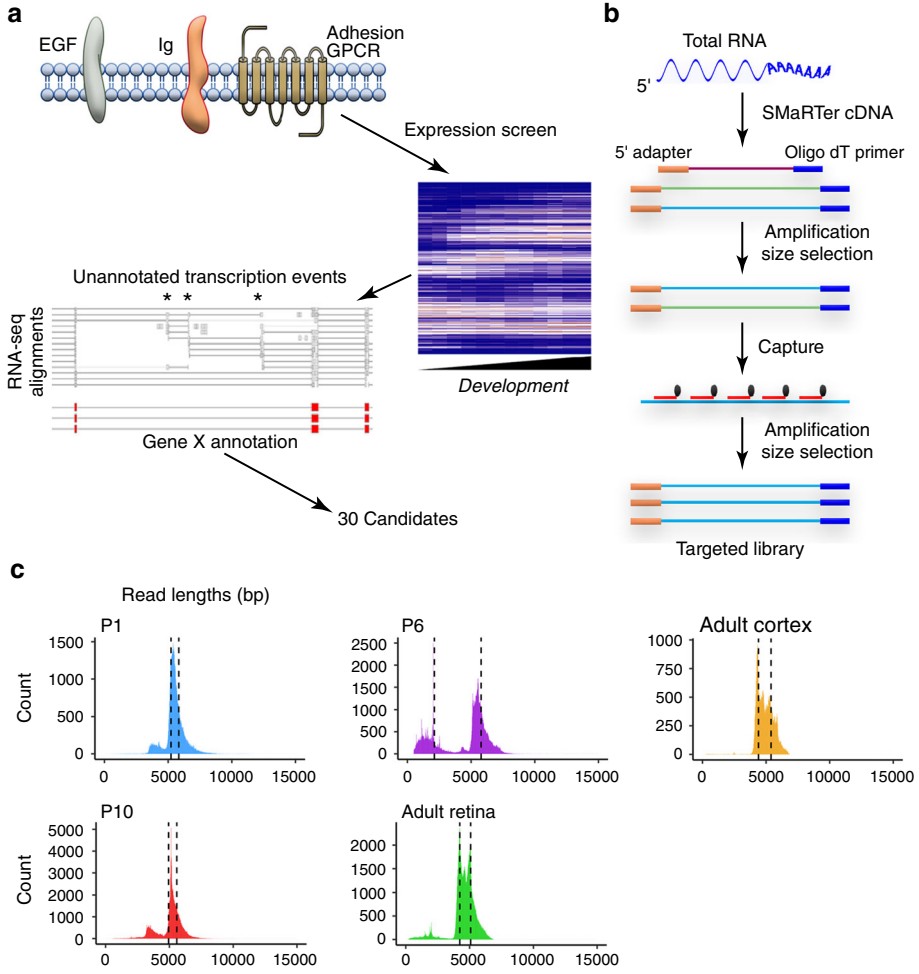

**Fig. 1 Strategy for identifying cell surface receptors that exhibit high isoform diversity. a** Screening strategy for selecting genes for lrCaptureSeq. Members of EGF, Ig, and adhesion GPCR families were tested for (1) expression during neural development, using RNA-seq data from retina and cortex; and (2) unannotated transcript diversity, based on RNA-seq read alignments to the UCSC Genes public database, which revealed use of unannotated exons, transcriptional start sites, and alternative splice sites. Thirty genes showing strong evidence for unannotated events (asterisks) were selected for targeted sequencing of full length transcripts (**b**, **c**). **b** lrCaptureSeq workflow. cDNAs are 5′ tagged to enable identification of full-length reads. Red, biotinylated capture probes tiling known exons. To obtain sequencing libraries enriched for intact cDNAs, two rounds of amplification and size selection were used. **c** Size distribution of full-length reads for each lrCaptureSeq experiment. Mouse retina or cortex transcripts were analyzed at the specified ages; adult mice were P35. The vast majority of reads are within expected size range for cDNAs of targeted genes. Dashed lines, quartiles of read length distribution.

To implement lrCaptureSeq, we first filtered the initial candidate list down to 30 that were predicted to encode cDNAs of similar length (4–8 kb). The final target list included genes involved in axon guidance, synaptogenesis, and neuron-glial interactions; it also included one gene, *Crb1*, which is implicated in inherited photoreceptor degeneration. Some targeted genes were known to generate many isoforms (*Nrxn1, Nrxn3*), but in most cases isoform diversity had not previously been characterized. When captured cDNAs were sequenced on the PacBio platform, ~132,000 full-length reads were generated per experiment (Supplementary Fig. 1c). These reads were strongly enriched for the targeted genes (Supplementary Fig. 1b), and the vast majority of reads were within the targeted length range (Fig. 1c). Thus, lrCaptureSeq can achieve deep full-length coverage of larger cDNAs that are underrepresented in other long-read datasets.

**A comprehensive isoform catalog generated by lrCaptureSeq.**
To catalog isoforms for all 30 genes across development and

across CNS regions, we performed lrCaptureSeq at a variety of timepoints in mouse retina and brain (Fig. 1c; Supplementary Fig. 1c). The number of isoforms, and reads comprising each, were determined using PacBio Iso-Seq software, together with custom software we developed for the analysis of isoform populations (IsoPops[41]). After this processing pipeline, the lrCaptureSeq catalog contained 4116 isoforms of the 30 targeted genes (Fig. 2a, b; Supplementary Data 1 and 2)—approximately one order of magnitude greater than the number of isoforms currently annotated for this gene set in public databases (Fig. 2b). It was also far higher than the number of isoforms predicted by popular short-read transcriptome assembly software (Supplementary Fig. 2a). Only 9% of lrCaptureSeq isoforms appeared in any of the databases we examined, suggesting most of them are novel.

To ensure that these unannotated isoforms are real, we used independent datasets to validate their transcription start sites and exon junctions. Cap analysis of gene expression[42] (CAGE) reads from adult mouse retina[43] corroborated 97.7% of transcription start sites identified by lrCaptureSeq (1051/1076 adult retina isoforms had CAGE-seq coverage at their 5′ end; Supplementary

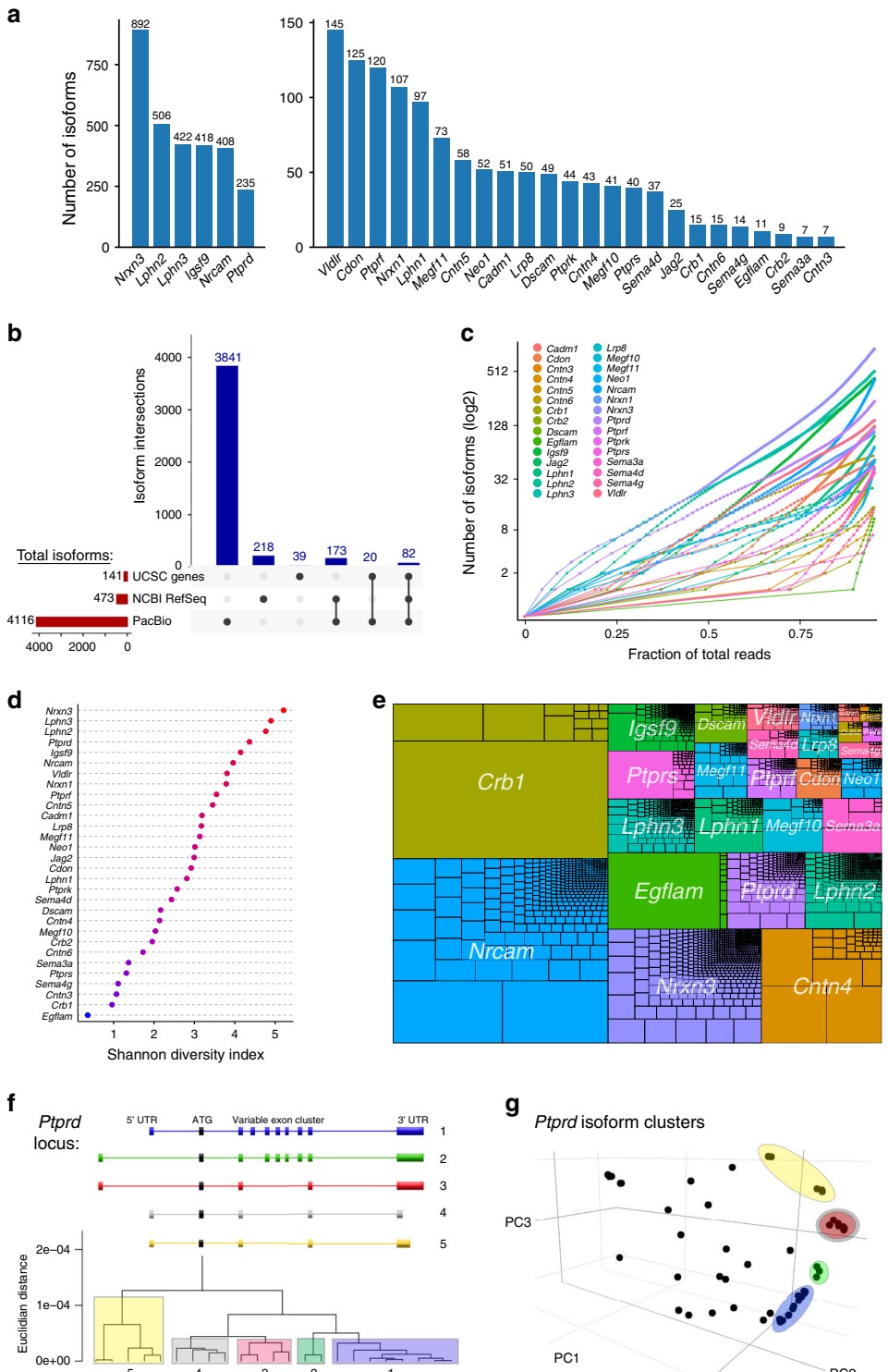

Fig. 1d). Moreover, CAGE-seq reads mapped selectively to 5′ ends of lrCaptureSeq isoforms (Supplementary Fig. 1d, e), further supporting the accuracy of our transcription start site annotations.

To validate lrCaptureSeq splice junctions we tested for their existence within short-read RNA-seq datasets from retina and brain[38,39]. Most lrCaptureSeq exon junctions (98.9%) occurred at canonical splice sites ($n = 80,590$ junctions), so we expected that this analysis would corroborate their validity. Indeed, we found that the independent datasets supported the vast majority (98.1%) of lrCaptureSeq junctions ($n = 79,020$). This included complete junction coverage for 71% of lrCaptureSeq isoforms ($n = 2925$). The unconfirmed junctions were likely absent from the RNA-seq data due to low expression levels, since the isoforms that did not show complete coverage were significantly less abundant (Supplementary Fig. 2b). Consistent with this interpretation, unconfirmed junctions could be detected by sequencing of RT-PCR products, suggesting that they were simply below RNA-seq detection threshold ($n = 9/12$ absent RNA-seq junctions in Megf11 gene were detected by RT-PCR). Together, these analyses strongly support the validity of the full-length sequences within the lrCaptureSeq isoform catalog.

**Fig. 2 mRNA isoform diversity revealed by lrCaptureSeq. a** Total number of isoforms cataloged for each gene after completion of lrCaptureSeq bioinformatic pipeline. **b** UpSet plot comparing isoform numbers in the PacBio lrCaptureSeq dataset with public databases (RefSeq, UCSC Genes). Intersections show that 53.9% of NCBI RefSeq isoforms were detected in the PacBio dataset (255 RefSeq isoforms, 4rd + 6th columns from left). For UCSC genes, 72.3% of isoforms annotated in this database were detected in the PacBio dataset (102 UCSC isoforms, 5th + 6th columns). **c** Lorenz plots depicting total number of isoforms cataloged for each gene (right Y intercepts), and fraction of each gene's total reads represented by each of its isoforms (dots). Curves are cumulative functions, with isoforms displayed in order from highest (left) to lowest (right) fraction of total gene reads. Also see Supplementary Fig. 2D. **d** Shannon diversity index was used to compare the relative diversity of each gene. Higher Shannon index reflects both higher isoform number and parity of isoform expression. **e** Treeplot depicting relative abundance of genes (colors) and isoforms (nested rectangles) within the entire dataset. Rectangle size is proportional to total read number. The most abundant isoform belonged to *Crb1*; the most abundant gene was *Nrcam*. **f**, **g** Unsupervised clustering applied at single gene level identifies families of related isoforms that share specific sequence elements. *Ptprd* gene is shown as an example. A subset of *Ptprd* isoforms cluster into 5 groups (F, bottom). These differ based upon 3 variables: length of 5′ UTR; length of 3′ UTR; and splicing of a variable exon cluster (**f**, top). The same groups segregate within principal components plot (**g**).

**Efficient isoform detection by lrCaptureSeq.** To probe the accuracy and sensitivity of isoform detection, we compared our lrCaptureSeq data to previous studies of the *Nrxn1* and *Nrxn3* genes. In these studies, the α and β classes of *Nrxn* transcripts were amplified by PCR and then characterized using PacBio sequencing[15,16]. The total number of *Nrxn1* and *Nrxn3* isoforms we identified was similar in scale to the previous studies (Fig. 2a), despite radically different library preparation and bioinformatic methods. Patterns of exon usage in alternative splice sites (AS)1–AS4 were also similar (Supplementary Table 1). For example, we confirmed a deterministic AS4 splicing event identified in the previous work, wherein *Nrxn3* exon 24 always splices to exon 25a ($n = 76$ exon 24-containing isoforms, all spliced to exon 25). These findings suggest that our *Nrxn* isoform catalog largely matches those generated by past studies. Nevertheless, we found features of the neurexin genes not noted in the previous catalogs. Because our method was not biased by PCR primer placement, we identified isoforms that did not contain canonical α or β transcript start/termination sites, one of which accounted for 64% of our *Nrxn3α* reads (Supplementary Table 1). Further, we detected 7 unannotated transcription termination sites, used by 16 different *Nrxn3α* isoforms, that truncate the mRNA upstream of the transmembrane domain (Supplementary Table 1). All seven of these new sites were corroborated with junction coverage from RNA-seq data. Together, these findings demonstrate the utility of lrCaptureSeq in recovering isoform diversity with high efficiency.

**Many isoforms contribute to overall gene expression.** Having identified a large number of isoforms within our lrCaptureSeq catalog, we next addressed whether this extensive isoform diversity is positioned to impact gene function. For diversity to be functionally significant, two conditions must be met: (1) multiple isoforms of individual genes should be expressed at meaningful levels; and (2) the sequences of the isoforms must differ enough to encode functional differences. To investigate isoform expression levels, we assessed how each gene's overall expression was distributed across its isoform portfolio (Fig. 2c, e; Supplementary Fig. 2d). Some genes—for example, *Egflam* and *Crb1*—were dominated by a small number of isoforms. However, other genes distributed their expression far more equitably across isoforms (Fig. 2c, e). Indeed, the genes that produced the largest number of isoforms also tended to be the most equitable, with a high fraction of similarly-abundant isoforms (Fig. 2c). Using the Shannon diversity index[44], we rank-ordered genes based on the diversity of their expressed mRNA species. *Nrnx3* was the top-ranked gene; however, several others of the latrophilin and protein tyrosine phosphatase receptor (PTPR) families scored nearly as high (Fig. 2d). Thus, *Nrxn3* is far from unique in expressing a large number of isoforms. We conclude that, for the genes in our

dataset, much of the isoform diversity is expressed at appreciable levels.

**Predicted functional diversity of lrCaptureSeq isoforms.** We next investigated the extent of sequence differences across the isoforms of each gene in our dataset. Most of the 30 genes encoded isoforms that varied widely in length and number of exons (Supplementary Fig. 2e, f), suggesting the potential for great functional diversity. To identify isoforms that are most likely to diverge functionally, unsupervised clustering methods were used to group isoforms based on their sequence similarity (Fig. 2f,g; Supplementary Fig. 2g). For most genes, isoforms clustered into distinct groups of related isoforms that made similar choices among alternative mRNA elements (Fig. 2f, g). Thus, major sequence differences exist within the isoform portfolio of individual genes, which can be traced to the inclusion of specific exon sequences by families of related isoforms.

To learn whether these sequence differences might diversify protein output, we analyzed predicted open reading frames (ORFs; Supplementary Data 1). The 4116 RNA isoforms in our dataset were predicted to express 2247 unique ORFs. A small subset of genes expressed great mRNA diversity but no equivalent ORF diversity (Fig. 3a); this was largely due to variations in 5′ UTRs or systematic intron retention (Supplementary Fig. 3c, d). Overall, however, there was a strong correlation between the number of isoforms and the number of predicted proteins (Fig. 3a). The amount of expressed ORF diversity varied by gene; but similar to mRNAs, a large amount of this predicted protein diversity was expressed at appreciable levels (Fig. 3b–d; Supplementary Fig. 3a, b). Remarkably, the genes with the most ORF diversity tended to encode a specific type of cell-surface protein: The top genes by Shannon diversity index all encode transsynaptic adhesion molecules (Fig. 3c). Thus, a major function of mRNA diversity may be the generation of protein variants that are positioned to influence formation or stability of synaptic connections.

To understand how mRNA diversity alters protein sequences, we studied the predicted protein output of individual genes. In many cases, predicted proteins varied substantially in their inclusion of well-characterized features or functional domains. This phenomenon is exemplified by the *Megf11* gene, which encodes a transmembrane EGF repeat protein implicated in cell-cell recognition during retinal development[45]. *Megf11* undergoes extensive alternative splicing: Out of 26 protein-coding exons, 21 are alternatively spliced (81%). In fact, we documented only ten constitutive splice junctions within the 234 *Megf11* isoforms identified in three independent long-read sequencing experiments (Fig. 4a, b; Supplementary Fig. 4). Examination of predicted proteins revealed a potential reason for such extensive splicing: Most of the EGF repeats comprising the extracellular domain are encoded by individual exons, such that alternative splicing causes

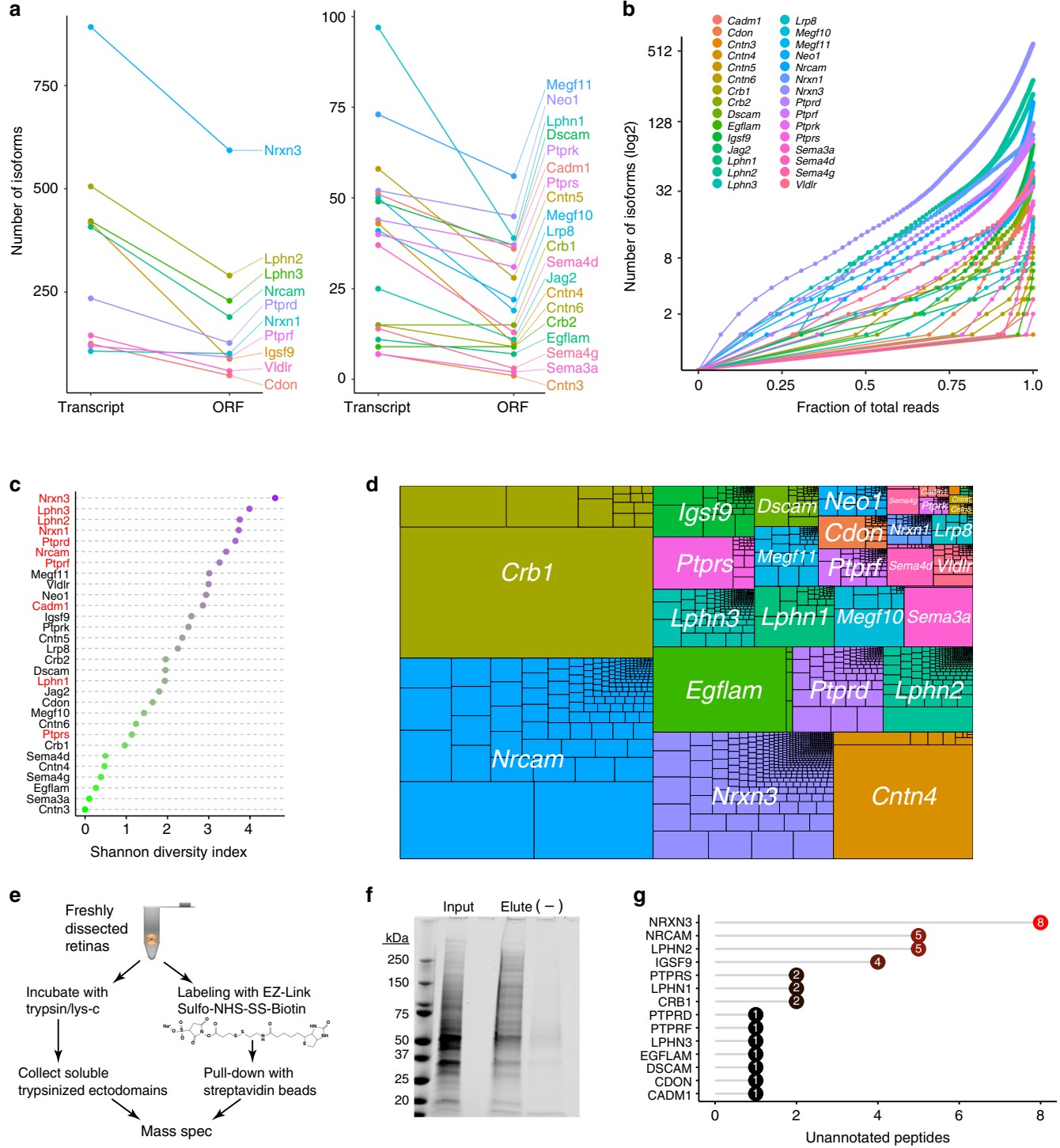

**Fig. 3 Transcript diversity contributes to a wealth of protein diversity. a** Total number of transcripts and ORFs for each gene in the lrCaptureSeq dataset. ORF number typically scales with transcript number, as shown by similar line slopes across most genes. A minority of genes exhibit far fewer ORFs than transcript isoforms (steep slopes). **b** Lorenz plots of isoform ORF distributions, similar to Fig. 2C. Many predicted protein isoforms (dots) are expected to contribute to overall gene expression. Also see Supplementary Fig. 3A, B). **c** Shannon diversity index for unique predicted ORFs for each gene. Genes that encode trans-synaptic binding proteins are highlighted in red. **d** Treeplot depicting relative abundance of predicted ORFs within the dataset. For most genes, overall expression is distributed across many ORF isoforms. Genes with steep slopes in **a** (e.g., *Cntn4*) show differences here compared with transcript treeplot (Fig. 2E). **e** Schematic of proteomic techniques used to enrich for cell surface proteins. **f** Coomasie stained protein gel from biotin-labeled and streptavidin-enriched cell surface proteins. Elution lane shows enrichment of higher molecular weight proteins compared with total lysate input (left lane). Bands from 75 to 250 kDa were excised for mass spectrometry. Right (−) lane, negative control sample omitting biotinylation reagent. **g** Plot depicting number of unannotated peptides discovered by mass spectrometry that do not exist in the UniProtKB database. Such peptides would have gone undetected if they had not been predicted to exist by lrCaptureSeq.

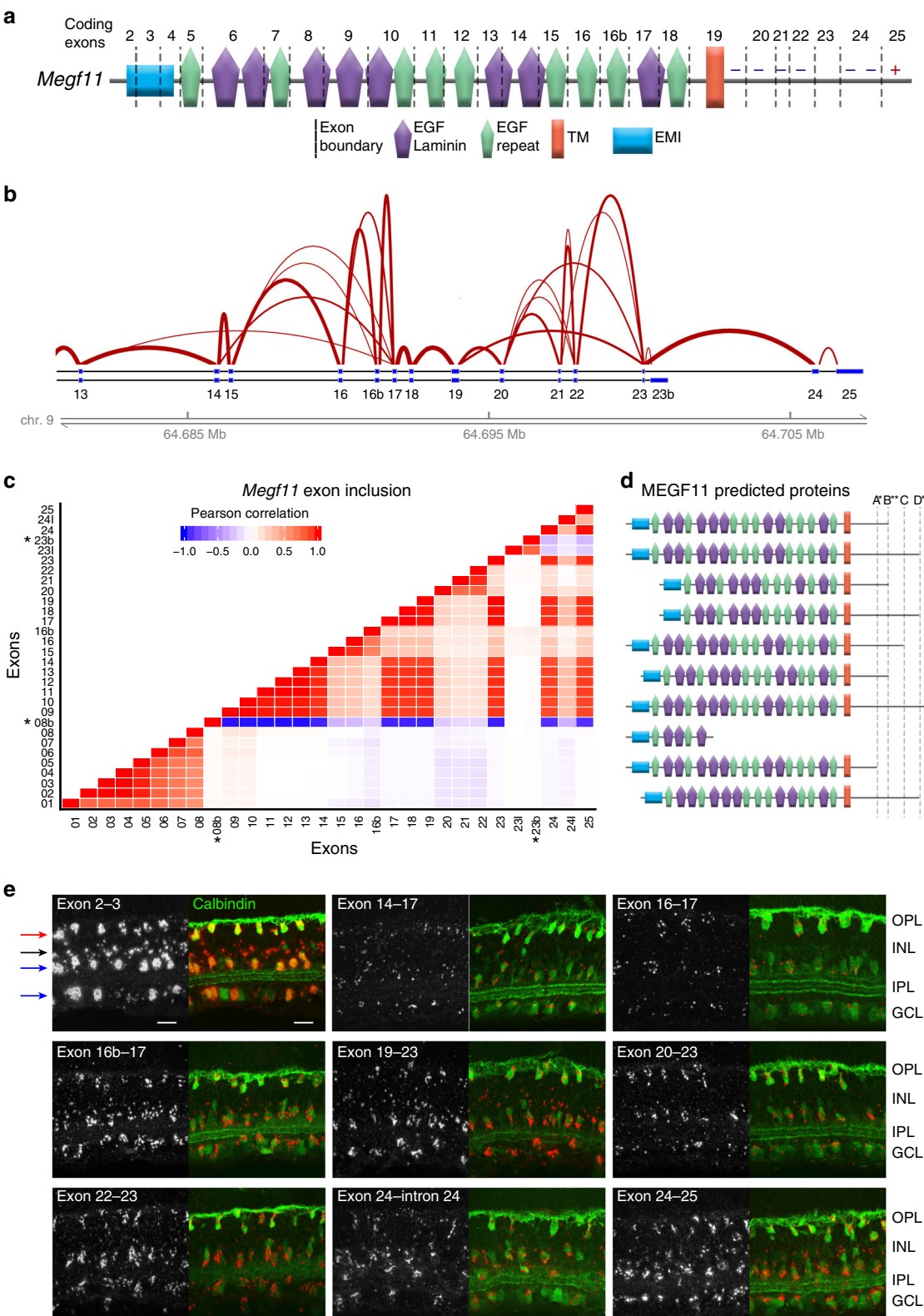

them to be deployed in a modular fashion (Fig. 4a–d). As a result of this modularity, predicted MEGF11 proteins showed substantial variability in the number and/or identity of included EGF repeats (Fig. 4d). The most variable EGF repeats were encoded by exons 14-16b (Fig. 4b); however, most of the EGF repeats were subject to alternative usage. Intracellular domain exons also showed potential for modularity in the use of ITAM or ITIM signaling motifs (Fig. 4a–d), similar to the situation in the Drosophila Megf11 homolog Draper[46]. Using BaseScope™ in situ hybridization[47,48], we confirmed that each of the most variable exon junctions are expressed by retinal neurons in vivo (Fig. 4e). Remarkably, individual Megf11-expressing cells used all of the exon junctions we tested, suggesting that extensive Megf11 isoform diversity is present even within individual neurons

**Fig. 4 Modular alternative splicing drives *Megf11* isoform diversity. a** Schematic of MEGF11 protein, showing how domain features correspond to exon boundaries. Individual EGF or EGF-Laminin (Lam) repeats are typically encoded by single exons. Splicing that truncates EGF-Lam domains (e.g., skipping of exon 14) is predicted to leave behind an intact EGF domain, preserving modularity. Putative intracellular domain signaling motifs: Plus sign, immunoreceptor tyrosine-based activation motif (YxxL/Ix$_{(6-8)}$YxxL/I); Minus sign, immunoreceptor tyrosine-based inhibitory motif (S/I/V/LxYxxI/V/L). TM, transmembrane domain; EMI, Emilin-homology domain. **b** *Megf11* sashimi plot generated from lrCaptureSeq data. Width of lines corresponds to frequency of splicing events. The most variable exon clusters (13–17 and 19–23) are shown. Exons in these clusters can splice to any downstream exon within the cluster. **c** Exon usage correlations across *Megf11* isoforms, calculated using IsoPops software. Positive correlation (red) is seen at short range among exons that show minimal splicing, e.g., 1-8 and 17-19. Long range correlations are largely absent, suggesting that most splicing is stochastic. Long-range negative correlations are only observed in the trivial case of exons downstream from an alternative transcription stop site (asterisks). **d** Predicted protein domains encoded by the 10 most abundant *Megf11* isoforms. Splicing varies number and identity of EGF domains on the extracellular portion of the protein, and produces 5 distinct cytoplasmic domains. Isoform 8 is the result of an alternative transcriptional stop site (**c**, exon 8b) and is predicted to encode a secreted isoform. Splicing from exon 19 to 20 (single asterisk) or retention of intron 24 (double asterisk) both result in frameshift and early stop codon. **e** BaseScope staining of P10 mouse retinal cross-sections, using probes targeting indicated splice junctions (red). Constitutive junction (2-3, top left) reveals full *Megf11* expression pattern, in four cell types: ON and OFF starburst amacrine cells (blue arrows), horizontal cells (red arrow), and an unidentified amacrine cell (black arrow). Calbindin (green) marks starburst and horizontal cells. Staining intensity for each junctional probe is consistent with junction frequency in sequencing data (see Sashimi plot, B). All junctions are expressed by all individual cells of the starburst and horizontal populations. Scale bar, 10 μm.

(Fig. 4e). Therefore, similar to insect *Dscam1*, *Megf11* uses alternative splicing of modular extracellular domain features to create a large family of isoforms encoding distinct cell-surface molecules. Together with our analysis of the full lrCaptureSeq dataset, these findings strongly suggest that isoform diversity serves to diversify the repertoire of neuronal cell-surface proteins.

**lrCaptureSeq isoforms encode cell-surface proteins in vivo**. To determine whether lrCaptureSeq isoforms are translated into proteins, we surveyed the retinal cell-surface proteome using mass spectrometry. Cell-surface protein samples were obtained from developing retina using cell-impermeant reagents that either cleaved or biotinylated extracellular epitopes (Fig. 3e, f). To learn whether these samples contained protein isoforms identified by lrCaptureSeq, we generated a database of possible trypsin peptide products derived from the isoforms within the lrCaptureSeq catalog. This was essential because protein identification requires comparison of raw mass spectrometry data to a reference peptide database. On generation of this predicted peptide database, we found that it contained ~25% more putative peptides for our 30 genes than the UniProt Mouse Reference Database typically used in most proteomics experiments (Supplementary Fig. 3e). The extra putative peptides represent the additional protein sequence complexity that is predicted by the lrCaptureSeq catalog.

Using this database as a reference, our mass spectrometry experiment identified 686 total peptides corresponding to 28 of the genes. 35 of these peptides were absent from the UniProt standard reference database, and were present only in our lrCaptureSeq reference (Fig. 3g; Supplementary Data 3). This fraction represents unannotated peptides, predicted from our lrCaptureSeq isoform catalog, that would have gone undetected in a typical mass spectrometry experiment. Unannotated peptides were found for 14 of our 30 genes, validating predicted exonic sequences, splice junctions, and splice acceptor sites (Supplementary Data 3). These findings strongly suggest that at least some of the proteins predicted by lrCaptureSeq are expressed on the surface of retinal cells in vivo.

**Identification of an abundant retina-specific *Crb1* isoform**. To investigate whether newly-discovered isoforms can provide insight into gene function, we focused on *Crb1*, a well-known retinal disease gene. Our *Crb1* catalog contained 15 isoforms, several of which were tissue-specific and developmentally-regulated (Fig. 5a, b; Supplementary Fig. 5b, c). In mature retina, *Crb1* expression was dominated by a single isoform—but

not the one that has been the subject of virtually all previous *Crb1* studies. Instead, the dominant isoform was a retina-specific variant bearing unique 5′ and 3′ exons (Figs. 5a, 6a; Supplementary Fig. 6a) and a unique putative promoter site just upstream of the 5′ exon (Fig. 5c). We named this isoform *Crb1-B*, to distinguish it from the canonical *Crb1-A* isoform.

Even though *Crb1-B* was the most abundant of the 4116 isoforms in our dataset (Fig. 2d), it was not annotated in the major genome databases (RefSeq, GENCODE, or UCSC). Nor, to our knowledge, was it documented in the literature. *CRB1-B* is also the most abundant isoform in human retina, as shown by a lrCaptureSeq dataset generated from human retinal cDNA (Figs. 5d and 6b). A third variant, *CRB1-C*, was also expressed in human retina at moderate levels—much higher than in mouse—but it was still not as abundant as *CRB1-B* (Figs. 5d and 6b). As in mouse, ATAC-seq revealed a putative *B* isoform promoter in human retina (Fig. 5c–e). Using short-read datasets[38,49], we corroborated the mouse and human findings (Fig. 6c, d) and extended them to several other vertebrate species (Supplementary Fig. 5a). Together, these results demonstrate that the major retinal isoform of an important disease gene had previously been overlooked: Across a range of vertebrate species, *CRB1-B* is the predominant *CRB1* isoform in the retina.

**Crb1 isoforms are expressed in different retinal cell types**. *Crb1-B* is predicted to encode a transmembrane protein sharing significant extracellular domain overlap with CRB1-A, but an entirely different intracellular domain (Fig. 7a, b). We therefore asked whether this protein is expressed and, if so, where the protein is localized. Western blotting with an antibody raised against the intracellular domain demonstrated that CRB1-B protein exists in vivo (Fig. 7c). Moreover, it exists in the configuration predicted by lrCaptureSeq (Fig. 7a), because intracellular domain expression was absent in mice engineered to lack the *Crb1-B* promoter and 5′ exon (Fig. 7c; see below for mouse design). Consistent with the notion that CRB1-B is a transmembrane protein, it was detected in the membrane fraction but not the soluble fraction of retinal lysates (Fig. 7d). Further, when expressed in heterologous cells, CRB1-B trafficked to the plasma membrane in a manner strongly resembling CRB1-A (Supplementary Fig. 6c). These data suggest that both major CRB1 isoforms localize at the cell surface.

To determine the expression patterns of *Crb1-A* and *Crb1-B*, we developed a strategy to evaluate expression of lrCaptureSeq isoforms within single-cell (sc) RNA-seq datasets. Applying this strategy to scRNA-seq data from developing mouse retina[50],

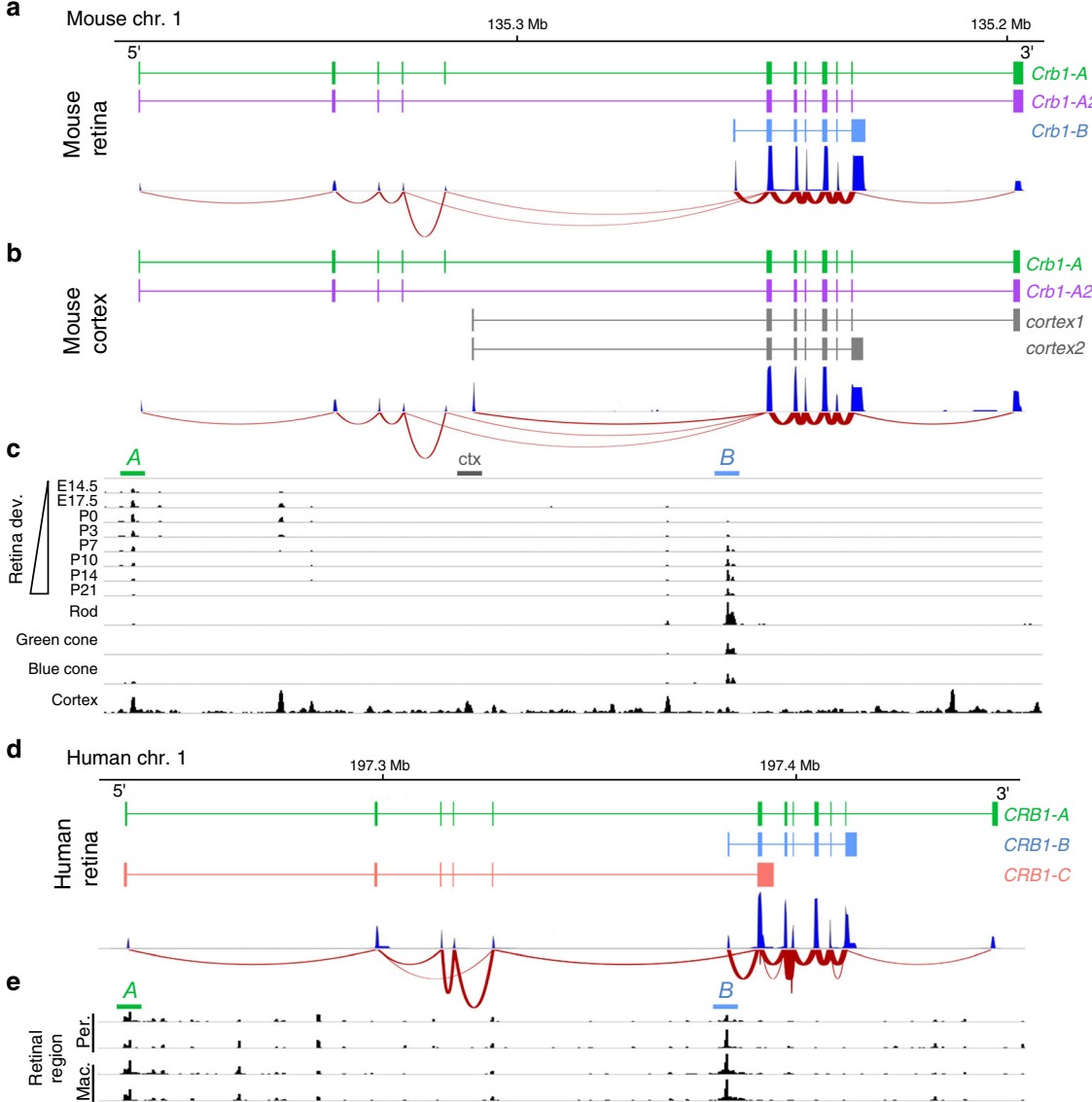

**Fig. 5 Identification of a retina-specific *Crb1* isoform in mouse and human retina.** Transcript maps of most abundant *Crb1* isoforms from mouse retina (**a**) and cortex (**b**). Map depicts reverse strand of mouse chromosome 1 at indicated location (Mb, megabase). 5′ and 3′ ends of *Crb1* transcripts are indicated. *A* is the canonical isoform; *A2* is a minor splice variant of *A*. These isoforms are shared between retina and cortex, whereas *Cortex 1*, *Cortex 2*, and *Crb1-B* are tissue-specific. Corresponding exon coverage (dark blue) and sashimi plots (red lines) were generated from lrCaptureSeq dataset. Note prevalence of reads associated with *Crb1-B* isoform (**a**). **c** Assay for chromatin accessibility (ATAC-seq; GSE102092, GSE83312) identifies likely promoters of *Crb1-A* and *-B* isoforms. Colored bars indicate location of putative *A* (green) and *B* (blue) promoters. Maps in **a–c** are aligned with each other. *Crb1-A* promoter is more open during development, but stays accessible in mature retina. *Crb1-B* promoter is open and presumed active in mature rods and both types of cones. Dnase I hypersensitivity data from ENCODE project reveals distinct chromatin environment in frontal cortex, consistent with expression of *A* isoform, as well as shorter cortex isoforms (*cortex 1* and *2*; gray bar at top). **d** Transcript maps of most abundant human retinal *CRB1* isoforms, identified by lrCaptureSeq. Map depicts forward strand of human chromosome 1 at indicated location. *A* and *B* isoforms are highly homologous to mouse (**a**). *CRB1-C* encodes a putative secreted form of the protein; it was also identified in the mouse dataset but its relative abundance in mouse was much lower than *A* and *B*. Note that *Crb1-A2* was not detected in the human dataset. Exon coverage (dark blue) and sashimi plots (red lines) were generated from lrCaptureSeq data. **e** ATAC-seq (GSE99287) of human peripheral (per.) and macular (mac.) retina show open regulatory sites corresponding to putative promoters for *CRB1-A* (green bar) and *CRB1-B* (blue bar). Two biological replicates are shown. Maps in **d**, **e** are aligned with each other.

we found distinct expression patterns for each isoform. *Crb1-A* was expressed largely by Müller glia (Fig. 8a, b; Supplementary Fig. 6d), consistent with previous immunohistochemical studies[37,51]. *Crb1-B*, by contrast, was expressed by rod and cone photoreceptors (Fig. 8a, b; Supplementary Fig. 6b, d). These cell-type-specific expression patterns were validated using two independent methods: First, ATAC-seq data from rods and cones showed that photoreceptors selectively use the *Crb1-B* promoter (Fig. 5c). Second, BaseScope staining confirmed

mutually exclusive expression of the two isoforms, with *Crb1-A* localizing to Müller cells and *Crb1-B* to photoreceptors (Fig. 8c).

To examine CRB1-B protein localization, we initially attempted immunohistochemistry but found that our antibody was not suitable. Therefore, we turned to a technique that combines serial tangential cryosectioning of the retina with Western blotting[52,53]. Each tangential section contains a specific subset of cellular and subcellular structures that are recognized by representative protein markers (Fig. 8d). This approach confirmed expression

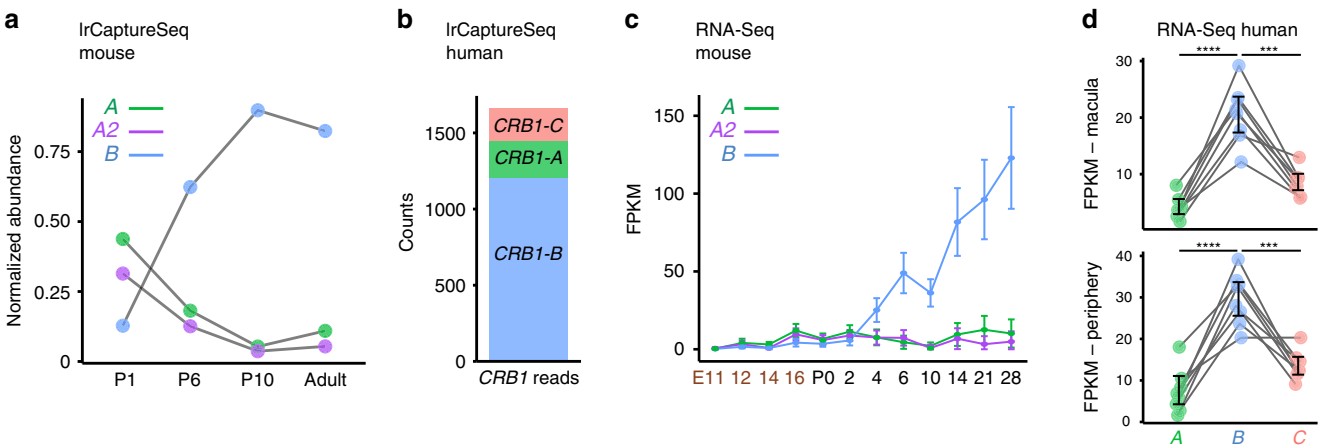

**Fig. 6 Crb1-B is the most abundant Crb1 isoform in mouse and human retina. a** Retinal expression of top 3 Crb1 isoforms across mouse development, quantified from lrCaptureSeq dataset. A isoforms predominate at P1 but Crb1-B becomes most abundant by P6. Data were normalized to total Crb1 read counts at each timepoint (P1 = 923 reads, P6 = 6127 reads, P10 = 14,007 reads, Adult = 10,975 reads). **b** Expression of top 3 human CRB1 isoforms, quantified from adult human retina lrCaptureSeq dataset. **c, d** Short-read RNA-seq data was used to quantify top 3 mouse (**c**) or human (**d**) CRB1 isoforms. Mouse dataset (GSE101986; n = 2 biological replicates per time point) confirms developmental regulation of each isoform observed in PacBio data (**a, c**). Human dataset (GSE94437; n = 8 biological replicates) confirms CRB1-B is dominant isoform in adult macula (**d**, top) and peripheral retina (**d**, bottom). Lines (**d**) show measurements derived from same donor. FPKM, fragments per kilobase of transcript per million mapped reads. Statistics (**d**): One-way ANOVA with Tukey's post-hoc test. ****$P < 1 \times 10^{-7}$. ***$P = 1.6 \times 10^{-6}$ (top); $P = 6.6 \times 10^{-6}$ (bottom). Error bars, 95% confidence interval of the FPKM value computed by Cufflinks software (**c**) or S.D. of the mean (**d**). For values, see Source Data file.

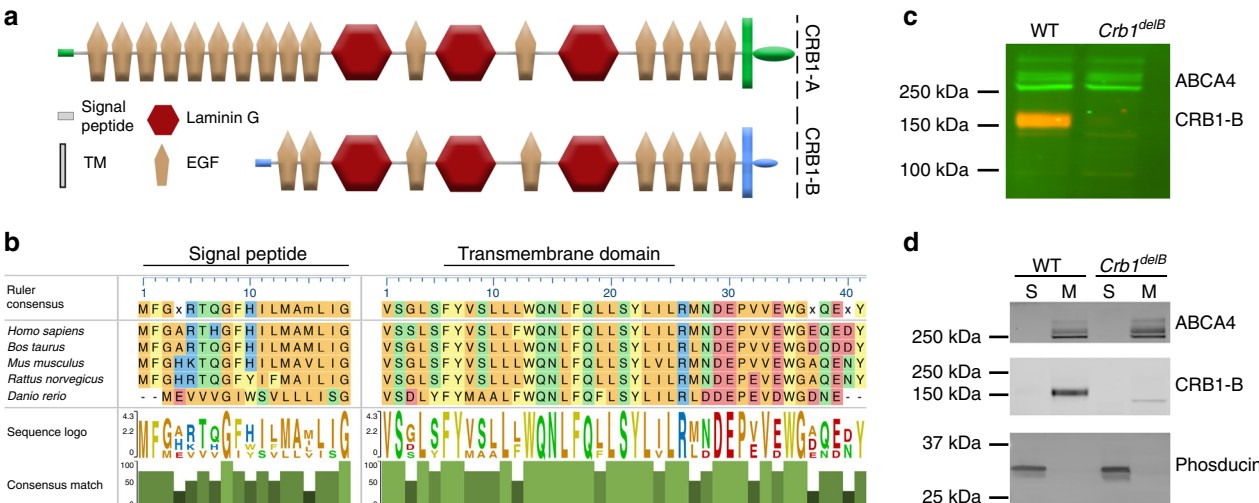

**Fig. 7 Characterization of CRB1-B protein. a** Domain structures of CRB1-A and CRB1-B protein isoforms. Green, A-specific regions; blue, B-specific regions. Each isoform has unique sequences at N-termini, predicted to encode signal peptides, and at C-termini, predicted to encode transmembrane (TM) and intracellular domains. **b** ClustalW alignment of unique CRB1-B sequences (blue in **a**). Both N- and C-terminal regions are conserved across vertebrate species. **c** Western blot demonstrating CRB1-B protein expression in retinal lysates. CRB1-B antibodies were generated against unique CRB1-B C-terminus. Deletion of Crb1-B first exon in mutant mice (Crb1[delB] allele; see Fig. 9a) demonstrates antibody specificity and also that the unique first and last exons of Crb1-B are primarily used together, as predicted at transcript level (Fig. 5a). Photoreceptor protein ABCA4 is used as loading control. Also see Source Data file. **d** Western blot on retinal lysates separated into soluble (S) and membrane-associated (M) protein fractions. CRB1-B is detected in the membrane fraction. Loading controls: Membrane fraction, ABCA4; soluble fraction, Phosducin. Also see Source Data file.

of CRB1-B in the photoreceptor layer, predominantly within the inner and outer segments. This localization is in marked contrast to CRB1-A which has been localized to the apical tips of Müller cells, within the OLM (Fig. 8a), using antibodies specific to this isoform[37,51].

**CRB1-B is required for outer limiting membrane integrity.** We next investigated the function of the CRB1-B isoform. Photoreceptors and Müller glia, the two cell types that express the major CRB1 isoforms (Fig. 8), engage in specialized cell-cell

junctions that form the OLM (Fig. 9b, c). It has been suggested that degenerative pathology in CRB1 disease may result from disruption of these junctions, but mouse studies have failed to clarify whether CRB1 is in fact required for OLM integrity. The two existing Crb1 mutant strains (Fig. 9a) have conflicting OLM phenotypes: Mice bearing a Crb1 point mutation known as rd8 show sporadic OLM disruptions[36], whereas a Crb1 "knockout" allele, here denoted Crb1[ex1], fails to disturb OLM junctions[37]. Our lrCaptureSeq data revealed a key difference between these two alleles: rd8 affects both Crb1-A and Crb1-B isoforms, whereas the "knockout" ex1 allele leaves Crb1-B intact (Fig. 9a). Therefore, we

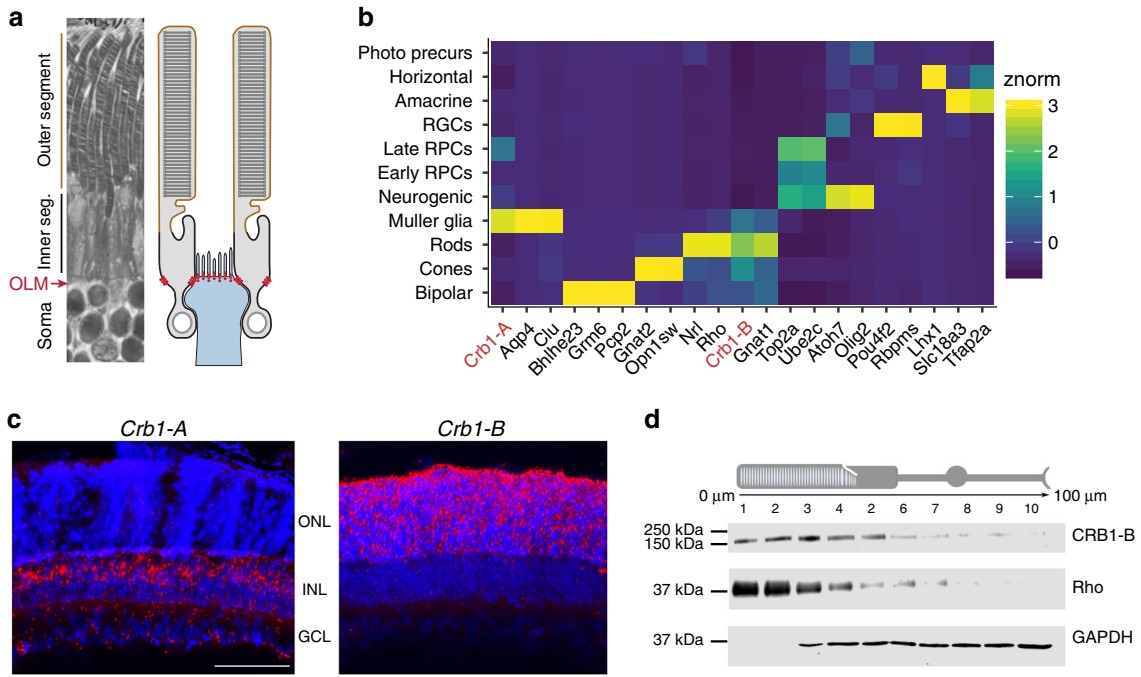

**Fig. 8 CRB1-B is expressed by photoreceptors. a** Schematic showing anatomy of outer retinal region where CRB1 is expressed. Left, photomicrograph depicting photoreceptor anatomy. Soma, inner segment (black), and outer segment (brown) regions are indicated. Outer limiting membrane (OLM; red arrow) separates soma layer from inner segment layer. Right, OLM anatomy schematic. OLM consists of junctions (red dots) between photoreceptors (gray) and Müller cells (blue). Junctions form selectively at particular subcellular domains of each cell type, i.e., glial apical membranes and photoreceptor inner segments. CRB1-A is expressed by Müller cells (**b-c**) where it localizes selectively to OLM junctions[51]. CRB1-B is expressed throughout the photoreceptor, including inner and outer segments (**b-d**). **b** Mapping of *Crb1* isoforms in scRNA-seq data[50]. Heat map generated from gene profiles of >90,000 cells, showing normalized expression of *Crb1* isoforms and retinal cell type marker genes. Unsupervised clustering was used to define genes co-expressed with *Crb1* isoforms. *Crb1-B* clusters with known cone and rod photoreceptor genes, while *Crb1-A* clusters with known Müller glia genes. **c** BaseScope staining of P20 mouse retina cross-sections using isoform-specific probes (red). Blue, Hoechst nuclear counterstain. *Crb1-A* probe targeted exon 1-2 junction, which is also used by *Crb1-A2* and *Crb1-C* (see Fig. 5A). Signal is primarily limited to central INL, where Müller cell bodies reside (left). *Crb1-B* probe targeted the junction between its unique 5' exon and exon 6 (Fig. 5A). Signal is limited to photoreceptors within ONL. *ONL* outer nuclear layer, *INL* inner nuclear layer, *GCL* ganglion cell layer. Scale bar, 100 µm. **d** Subcellular localization of CRB1-B within rod photoreceptors, assessed by Western blotting of serial 10 µm tangential sections through mouse outer retina. Each lane corresponds to photoreceptor cellular compartment denoted by cartoon at top. Rhodopsin (Rho, center) is an outer segment marker; GAPDH (bottom) is excluded from outer segment but is present throughout the rest of the cell. CRB1-B protein (top) is present in all compartments; expression is strongest in lanes corresponding to outer and inner segments. Also see Source Data file.

hypothesized that *Crb1-B* influences the integrity of photoreceptor-Müller junctions at the OLM. To test this hypothesis, we generated two new mutant alleles (Fig. 9a; Supplementary Fig. 7a, b). The first, *Crb1^delB^*, abolishes *Crb1-B* while preserving other isoforms including *Crb1-A*. The second, *Crb1^null^*, is a large deletion designed to disrupt all *Crb1* isoforms.

Using electron microscopy to evaluate OLM integrity, we found that *Crb1^null^* mutants exhibit disruptions at the OLM whereby photoreceptor nuclei invaded the inner segment layer, disturbing the structure of the outer retina (Fig. 9b–e; Supplementary Fig. 7d). Within the disrupted regions, photoreceptor inner segments lacked their characteristic electron-dense junctions with apical Müller processes, indicating that OLM gaps arose due to disruption of photoreceptor-Müller contacts (Fig. 9f). A similar phenotype was also observed in *Crb1^rd8^* mutants, as previously reported[36] (Fig. 9f, g, j; Supplementary Fig. 7d–f). To explore the contribution of each isoform to the OLM phenotype, we examined mice bearing various combinations of the *Crb1^null^* and *Crb1^delB^* alleles. In *Crb1^delB/delB^* mice, which lack *Crb1-B* but retain two copies of *Crb1-A*, the OLM phenotype was still evident but was weaker than in *rd8* or *null* homozygotes (Fig. 9h, j). By contrast, the OLM phenotype was equivalent to *rd8* and *null* mutants in *Crb1^delB/null^* mice, which lack *Crb1-B* but retain one copy of *Crb1-A* (Fig. 9e, j; Supplementary Fig. 7f). These findings

indicate that both *Crb1* isoforms are needed for OLM junctional integrity, but the role of *Crb1-B* is particularly important, given that OLM disruptions can arise even when *Crb1-A* remains present.

**Retinal degeneration in mice lacking all *Crb1* isoforms.** Finally, we asked whether insight into CRB1 isoforms could be used to improve animal models of CRB1 degenerative disease. Photoreceptor degeneration is absent or extremely slow in existing *Crb1* mutant mice, making them poor models of human degenerative phenotypes[36,37,54]. We hypothesized that previously unannotated *Crb1* isoforms, such as *Crb1-B*, might help explain these mild phenotypes. Consistent with this possibility, we noted that neither *Crb1^ex1^* nor *Crb1^rd8^* completely eliminates all *Crb1* isoforms (Fig. 9a). To test the contribution of new *Crb1* isoforms to photoreceptor degeneration, we took advantage of our newly-generated *Crb1^delB^* and *Crb1^null^* strains (Fig. 9a). Quantification of photoreceptor numbers in young adult mice (P100) revealed that both *Crb1-A* and *Crb1-B* isoforms are required for photoreceptor survival. *Crb1^delB^* mutants had normal photoreceptor numbers (Fig. 10a, d; Supplementary Fig. 7c), similar to the previously-reported *Crb1^ex1^* mutant[37]. Therefore, removing either major isoform by itself has minimal degenerative effects. By contrast, deletion of all isoforms in *Crb1^null^* mice caused marked

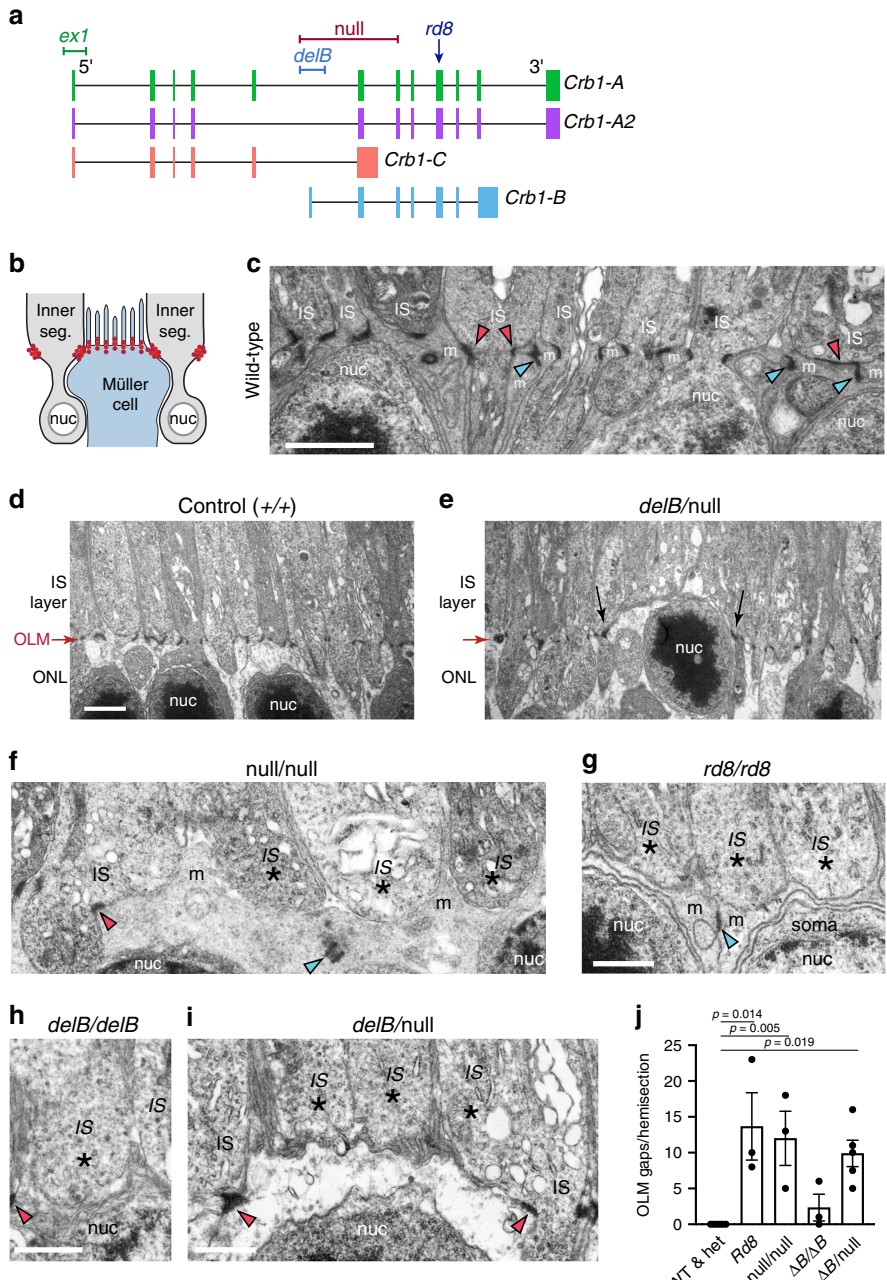

**Fig. 9 Requirement for *Crb1* isoforms in outer limiting membrane integrity. a** Schematic of *Crb1* locus showing genetic lesions underlying mouse mutant alleles. Previously studied alleles: *Crb1ex1*, a targeted deletion of exon 1 that does not impact *Crb1-B*; *Crb1rd8*, a point mutation in exon 9. Alleles generated here: *Crb1delB*, a CRISPR-mediated deletion of the first *Crb1-B* exon and its promoter region, leaving *Crb1-A* intact; *Crb1null*, a CRISPR-mediated deletion of exons used in all *Crb1* isoforms. Also see Supplementary Fig. 7A. **b** Schematic illustrating location of OLM junctions (red) surrounding photoreceptor inner segments. **c** Electron micrograph from wild-type mouse. All inner segments make OLM junctions with Müller cells. *IS* inner segment. Red arrowheads, photoreceptor-glial junctions. Blue arrowheads, glial-glial junctions. **d**, **e** OLM phenotype in *Crb1* mutants. In wild-type control retina (**d**), OLM (red arrow) divides outer nuclear layer (ONL) from IS layer. In *Crb1* mutants (**e**), OLM gaps allow nuclei to penetrate IS layer. Arrows demarcate region lacking OLM junctions. Image depicts *Crb1delB/null* mutant, but is representative of OLM phenotypes in *null*, *delB*, and *rd8* mutants (Supplementary Fig. 7D-F). **f–i** Higher power views of OLM gaps in *Crb1* mutants. In each allelic combination, inner segments lacking OLM junctions (asterisks) were observed. Red and blue arrowheads as in **c**. **j** Quantification of OLM gap frequency. No gaps were observed in wild-type (WT) or *Crb1null/+* heterozygote (het) controls. Gap frequency was similar in *rd8*, *null*, and *delB/null* mutants, the latter of which lack *Crb1-B* but still express *Crb1-A*. Statistics, one-way ANOVA with Tukey's post-hoc test. *Null*, *rd8*, and *delB/null* differed significantly from controls (*P*-values given on graph), but did not differ significantly from each other (*rd8* vs. *null* P = 0.991; *rd8* vs. *delB/null* P = 0.784; *null* vs. *delB/null* P = 0.967). Also see Supplementary Fig. 7F and Source Data file. Sample sizes: n = 3 (*null/+*, *rd8*; *null/null*, *delB/delB*); n = 5 (*null/delB*); n = 6 (WT). WT and het were pooled for plotting and statistics. Error bars, S.E.M. Scale bars 2 μm (C,-E); 1 μm (F-I). Bar in D applies to E; bar in G applies to F.

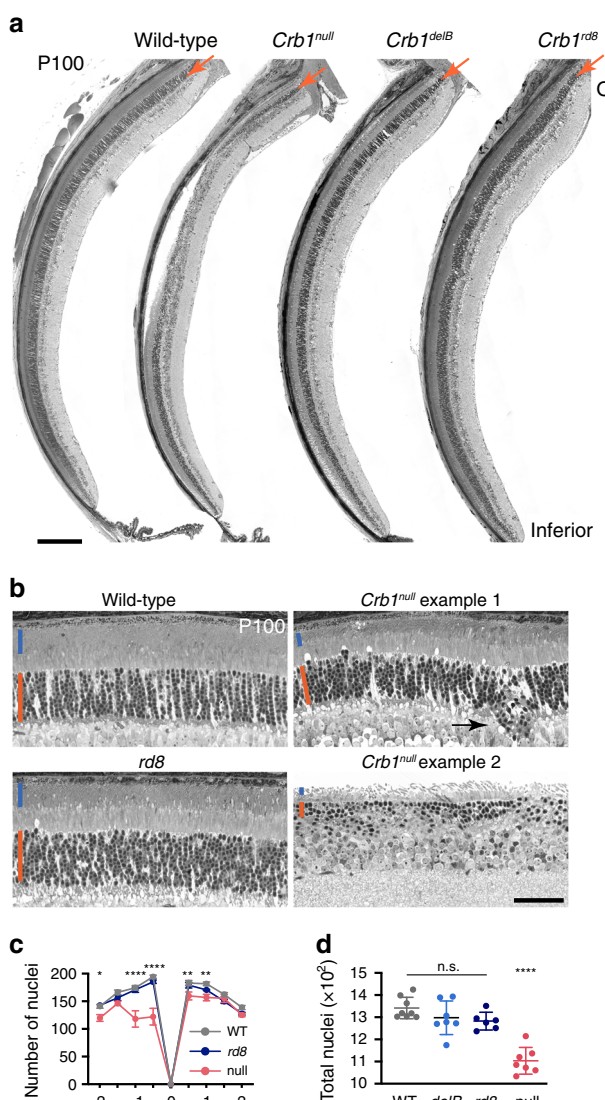

**Fig. 10 Ablation of all *Crb1* isoforms causes retinal degeneration. a** Retinal histology in *Crb1* mutant mice at P100. Thin plastic sections through inferior hemisphere are shown for homozygous mutants of indicated genotype, and wild-type controls. Arrow, ONL layer containing photoreceptor nuclei. A particularly severely affected *Crb1null* retina is shown; note large focal region of photoreceptor loss accompanied by retinal detachment. Areas outside the most aggressively degenerative patch show ONL thinning. *Crb1delB* and *Crb1rd8* mutants show no apparent loss of ONL cells. ONH, optic nerve head. **b** Higher magnification views of retinal histology, 450 μm inferior to ONH. Images from two different *Crb1null* animals are shown to highlight variability in focal degeneration. Even the mild null case has thinner ONL (orange line) with fewer nuclei than age-matched *Crb1rd8*. Outer segment length (blue line) is also diminished in null mutants. **c**, **d** Quantification of ONL cell number at P100. Spider plot (**c**) shows counts of ONL nuclei in 100 μm bins distributed uniformly across retinal sections (e.g., **a**). Left, inferior side. For *Crb1delB* spider plot see Supplementary Fig. 7. **d** Total ONL nuclei counted in all 8 bins. Statistics (**c**): two-way ANOVA with Sidak post-hoc test. *P* values refer to WT vs. null comparison; *rd8* was not significantly different from WT at any location. *$P = 0.015$; **$P = 0.007$, $P = 0.004$; ****$P < 1 \times 10^{-7}$. Statistics (**d**): one-way ANOVA with Tukey's post-hoc test. *Crb1null* was significantly different from all other groups. ****WT vs. null, $P = 2 \times 10^{-7}$; *rd8* vs. null, $P = 5.7 \times 10^{-5}$; delB vs. null, $P = 1 \times 10^{-5}$. None of the other *Crb1* mutants differed from WT or each other. Sample sizes: $n = 8$ WT; $n = 7$ null mutants; $n = 7$ delB mutants; $n = 6$ *rd8* mutants. For values see Source Data file. Error bars, S. E. M. Scale bars: 200 μm (**a**); 50 μm (**b**).

photoreceptor degeneration (Fig. 10a–d). Thus, significant cell loss requires compromise of both *Crb1-A* and *Crb1-B*. No degeneration was evident yet at P100 in *Crb1rd8* mutants (Fig. 10b–d), consistent with previous reports that significant degeneration takes ~2 years[36,54,55]. Together, these genetic experiments support the conclusion that multiple *Crb1* isoforms contribute to photoreceptor survival—including *Crb1-B*. Thus, modeling of human disease can be achieved by rational design of mutant alleles guided by lrCaptureSeq isoform catalogs.

## Discussion

Despite recent advances in sequencing technology, the true diversity of the transcriptome remains murky[12]. For most genes, only a small subset of the full isoform portfolio has been documented. Here we show that lrCaptureSeq can unveil the rich diversity of the CNS transcriptome. LrCaptureSeq is accurate and efficient, with sufficient depth to reveal the full-length sequence of even low-abundance isoforms. To facilitate interpretation of lrCaptureSeq data we provide a companion R software package for analyzing and visualizing isoform catalogs. Applying these tools to the developing nervous system, we uncovered a vast diversity of isoforms encoding cell surface proteins, most of which were unannotated elsewhere. Many were predicted to alter

functional protein domains. Further, we found that *Crb1-B*, the most abundant isoform in our entire dataset, has a distinct expression pattern and function from the canonical *Crb1* isoform, endowing it with disease-relevant functions. CRB1 therefore serves as a striking example of the value of comprehensive full-length isoform identification. Application of lrCaptureSeq to other cell types and tissues has great potential to unlock many new insights into gene function and dysfunction—both in the CNS and beyond.

LrCaptureSeq is successful because it enables deep long-read sequencing for transcripts that would be poorly represented in existing PacBio transcriptomes, due to their cDNA size and expression levels. Even with short-read sequencing, it is challenging to achieve sufficient sequencing depth for isoform identification[31,56]. Targeted CaptureSeq approaches improve short-read detection of low-abundance transcripts[31,57]; here we show the same is true for full-length sequencing of protein-coding mRNAs. It is clear from the distribution of isoform abundances (Fig. 2c) that only the least abundant isoforms escaped detection. Therefore, we consider the lrCaptureSeq isoform catalogs to be largely complete. However, we cannot exclude the possibility that certain transcripts are missing from the catalogs. For example, some isoforms smaller than 4.5 kb may have evaded detection due to the size selection step of our library preparation protocol (Fig. 1b). We suspect this is a small minority of transcripts because, even with size selection, we still cataloged numerous smaller isoforms (Supplementary Fig. 2e)—including *Crb1-B* (3.0 kb). Thus, while the catalogs may lack certain short and/or rare transcripts, we conclude that we have detected most of the isoforms expressed in our targeted tissues. We achieved this depth by targeting 30 genes for parallel sequencing, but higher-throughput PacBio instruments are now available; these should allow substantially more targeted genes to be sequenced in parallel without sacrificing isoform coverage.

Our results suggest many potential uses for lrCaptureSeq in transcriptome annotation. One particularly exciting use case is identification of cell-type-specific isoform expression patterns.

We show that lrCaptureSeq data can be integrated with existing short-read RNA-seq datasets, including single-cell data, to reveal the time and place of isoform expression. As of now, this approach works best for isoforms that differ at their 3′ ends, due to 3′ bias inherent in most single-cell library preparation methods. As scRNA-seq methods are refined to improve depth and coverage, we expect that other types of isoforms will become amenable to mapping in this way. With this methodology, it will not be necessary to generate lrCaptureSeq catalogs for each cell type in the nervous system; rather, cell-type-specific isoform expression can be determined bioinformatically by combining different types of sequencing data.

How many mRNA isoforms are produced by any given gene? While our 30 genes probably have more isoforms than the average gene, given that they were selected because they showed evidence of transcript diversity (Fig. 1a), our results suggest that the number is substantially higher than suggested by present database annotations. For the 30 genes in our dataset the median number of RefSeq isoforms was 11.5, and no gene had more than 51. By contrast, the median number of isoforms in our lrCaptureSeq catalog was 50, while the most diverse gene, *Nrxn3*, had nearly 900. A previous CaptureSeq study of long noncoding RNAs found only two-fold more isoforms[40], but this study was not focused on the nervous system. Thus, it remains to be determined whether the diversity we observed is a specific feature of CNS cell surface molecules, or if instead it is typical of other gene classes and tissues. Broader application of lrCaptureSeq should help resolve this question.

It has long been suspected that extensive cell-surface protein diversity might mediate formation of precise neuronal connections[5,8,58]. However, the need for numerous cell-surface cues has recently been called into question[59]. Here we show that extensive isoform diversity is widespread across many cell-surface receptor genes, and that individual neurons most likely express numerous of isoforms of certain genes (e.g., *Megf11*; Fig. 4e). Furthermore, our *Megf11* results show that insect Dscam1 is not unique in using large-scale modular alternative splicing to swap extracellular domain motifs. The phenomenon of modular EGF-repeat swapping through alternative splicing has been observed before, albeit at smaller scale, for Netrin-G proteins[60]. Therefore, other EGF-repeat genes may also generate large families of cell surface proteins using a similar strategy. Overall, our results establish the molecular prerequisite for models of CNS wiring that require numerous cell surface cues. Whether such models are correct remains to be determined; however, in this regard it is striking that the genes with the most predicted protein diversity share a common function as trans-synaptic, synaptogenic cell adhesion molecules[19,61,62] (Fig. 3c). Therefore, these diverse molecular cues are likely positioned in exactly the right place to influence the precision of synaptic connections.

Our studies of CRB1 illustrate the value and importance of documenting the complete isoform output of individual genes. CRB1 is a major causal gene for inherited retinal degenerative diseases, including Leber's congenital amaurosis, retinitis pigmentosa, and macular dystrophy[63–65]. As such, both mouse *Crb1* and human *CRB1* have been studied intensively. *CRB1-B* may have been overlooked because its 5′ and 3′ exons are the only parts of the transcript that distinguish it from *CRB1-A*. With short-read sequencing it is difficult to tell that these two distant exons are typically used together in the same transcript. By contrast, lrCaptureSeq clearly showed that the most abundant retinal *CRB1* isoform was a variant containing these unconventional 5′ and 3′ exons.

Due to their distinct 5′ and 3′ ends (Fig. 7a), *Crb1-A* and *-B* differ in crucial ways that likely endow them with distinct functions. Their 5′ exons have different promoters that drive

expression in different cell types—*Crb1-A* in Müller glia and *Crb1-B* in photoreceptors—while their 3′ exons encode different intracellular domains. The CRB1-A intracellular domain, like other vertebrate homologs of Drosophila Crumbs, contains two highly-conserved motifs mediating interactions with polarity proteins[66]. These motifs localize Crumbs homologs to apical junctions, where they are required for maintaining epithelial structural integrity and apico-basal polarity[33]. CRB1-B lacks these conserved motifs, suggesting a model whereby CRB1-A and -B operate in different cell types through different intracellular interaction partners.

Our findings have implications for the prevailing model of CRB1 disease, which posits that CRB1 is required for integrity of OLM junctions between Müller glia and photoreceptors[26]. A major challenge for this model has been the absence of OLM phenotypes or photoreceptor degeneration in *Crb1ex1* mutants, which lack only *Crb1-A* (Fig. 9a) but was thought to be a null allele[37]. As such, the weak mutant phenotype suggested that CRB1 might be dispensable for photoreceptor survival in mice[26]. Here we show that CRB1 is indeed required for OLM integrity and photoreceptor survival, but the mechanism involves the photoreceptor-specific CRB1-B isoform. Moreover, we show a genetic interaction between the two isoforms, revealing OLM integrity and pro-survival functions for CRB1-A that were obscured in *Crb1ex1*. We propose that the concerted action of CRB1-A in glia and CRB1-B in photoreceptors controls OLM integrity and photoreceptor health, perhaps through the assembly or maintenance of the junctional protein complex in each respective cell type.

The notion of concerted *Crb1-A* and *Crb1-B* function is further supported by the fact that *Crb1rd8* (Fig. 9a) has a more severe phenotype than *Crb1ex1*[36,37,55]. However, it remains unclear why *Crb1rd8* is less severe than *Crb1null* (Fig. 10), even though *A* and *B* isoforms are affected in both mutants. One possible explanation is that *Crb1rd8* may not be a mRNA or protein null[55]. Another possibility is that the *Crb1-C* isoform may play a compensatory role, as it is unaffected by *Crb1rd8* (Fig. 9a). Either way, our results show that the design of mouse disease models is significantly enhanced when a complete isoform catalog is available.

Overall, our work highlights the value of building complete and accurate full-length isoform catalogs. Lack of such information can cause key gene functions to be overlooked and can lead to misinterpretation of genetic experiments and disease phenotypes. We expect the transcriptomic ground truth provided by deep long-read capture sequencing will be an important addition to the transcriptome annotation toolbox, enabling discovery of specific mRNA isoforms that contribute to a wide range of normal and disease processes.

## Methods

**Resources and reagents**. All key reagents used in this study, including antibodies, primers, datasets, and animal strains, are listed in a Key Resources table (Supplementary Table 2).

**Animals and human tissue samples**. Mouse experiments in this study were approved by the Duke University Institutional Animal Care and Use Committee (protocols A005-16-01 and A274-18-12). The mice were housed under a 12 h light-dark cycle with ad lib access to food and water. Heat and humidity were maintained within the parameters specified in the National Institute of Health Guide for the Care and Use of Laboratory Animals. Experimental procedures were also consistent with this Guide.

Human donor eyes were obtained from Miracles in Sight (Winston Salem, NC), which were distributed by BioSight (Duke University Shared Resource). Ethical procedures, including procedures for obtaining informed consent from donors, were reviewed and approved by the Duke University Institutional Review Board (protocol #PRO-00050810). Postmortem human donor eyes were enucleated and stored on ice in PBS until dissection. Retinas were dissected from posterior poles and proceeded to RNA isolation. Donors with a history of retinal disease were excluded from the study.

**Knockout mouse generation**. For the generation of $Crb1^{delB}$, CRISPR guides were designed to target genomic coordinates chr1:139,256,486 and 139,254,837 and validated in vitro on genomic DNA prior to injection. A C57Bl6J/SJL F1 hybrid mouse line was used for injection; both strains are wild-type at the $Crb1$ locus (i.e., they do not carry $rd8$). Founders were genotyped using PCR primers to distinguish the alleles (see Supplementary Table 2 for primer sequences). Two founder lines with genomic deletions were maintained. One carrying the deletion 139,254,836–139,256,488 (Δ1652 bp) plus two additional cytosines, and the other 139,254,836–139,256,488 (Δ1652 bp). Both alleles effectively delete the entire first exon of $Crb1$-$B$ and the promoter region and are currently phenotypically indistinguishable. For the generation of $Crb1^{null}$, CRISPR guides were designed to target genomic coordinates chr1:139,256,486–139,243,407 and validated in vitro on genomic DNA prior to injection. A C57Bl6J/SJL F1 hybrid mouse line was used for injection and founders were genotyped using PCR primers (Supplementary Table 2) to distinguish the alleles. Two founder lines with genomic deletions were maintained. One carrying the deletion chr1: 139,256,844–139,243,411 (Δ13,433 bp) and the other 139,257,194–139,243,411 (Δ13,783 bp). Both alleles effectively delete the entire first exon of $Crb1$-$B$ and the promoter region in addition to exon 6 and part of exon 7 of $Crb1$-$A$. This deletion would eliminate the exon 7 splice acceptor and is predicted to exclude exon 7 altogether. Splicing from exons 5 to 8 (as in $Crb1$-$A$) and 4 to 8 (as in $Crb1$-$A2$) would result in frameshifts. The $Crb1$-$C$-specific retained intron after exon 6 is also entirely deleted. Founder animals were backcrossed with C57Bl6J mice for at least two generations before analysis and genotyped to ensure they were not carrying $rd1$ mutation from the SJL background. Animals generated in this study will be made available to the research community for non-commercial use.

**CRB1-B antibody**. We used Pierce Custom antibody service (Thermo Fisher Scientific) to generate a CRB1-B specific antibody. The antigen was the last 16 amino acids (RMNDEPVVEWGAQENY) of CRB1-B, which are predicted to be exclusive to this isoform at the protein level. Antibodies were made in rabbit according to their 90-day protocol with initial inoculation followed by 3 boosts. The antibody was affinity-purified and validated by western blot with a $Crb1^{delB}$ knockout control. CRB1-B produces a band of ~150 kDa, larger than the predicted size of 110 kDa. This discrepancy in experimental vs predicted size is likely due to post translational modifications such as glycosylation, since addition of PNGase F lowered the band size. Antibodies generated in this study will be made available to the research community for non-commercial use.

**RNA extraction**. For PacBio sequencing experiments and qRT-PCR, C57Bl6/J mice were anesthetized at P1, P6, P10, or P35 (adult) with isoflurane or cryoanesthesia (neonates only) followed by decapitation. Eyes were enucleated and retinas were dissected out, or brain was dissected from the skull and the cerebral cortex was removed. Total RNA was isolated using Tri Reagent (ThermoFisher Scientific AM9738) according to the manufacturer's protocol. Tissue was mechanically homogenized in Tri Reagent followed by phase separation with chloroform and isopropanol precipitation. RNA samples were stored at −80 °C. RIN number was calculated using a Bioanalyzer. Only RIN values above 9 were used for sequencing.

**PacBio library preparation for mouse samples**. Reverse transcription was carried out using the Clontech SMARTer cDNA kit according to the manufacturer's protocol. cDNA was amplified with KAPA HiFi DNA Polymerase for 12 cycles followed by size selection (4.5–10 Kb). For capture, 1 ug of cDNA was denatured and blocked with DTT primer and Clontech primer then mixed with Nimblegen's SeqCap EZ Developer (≤200 Mb) custom baits at 47 °C for 20 h. Biotynaylated cDNAs were pulled down with streptavidin beads and washed with Nimblegen hybridization buffers to minimize non-specific binding. Targeted cDNA library was amplified 11 cycles with Takara LA Taq. SMRT bell library was constructed then additional size selection (4.5–10 Kb) followed by binding of Polymerase with P6-C4 chemistry (RSII). Library was loaded onto SMRT cell using MagBead loading at 80pM (RSII). For PacBio Sequel library, sequencing primer version 2.1 was annealed and bound using polymerase version 2.0. The bound complex was cleaned with PB Ampure beads and loaded by diffusion at 6 pM with 120 min pre-extension.

**PacBio library prep for human retina**. Reverse transcription was carried out using Clontech SMARTer cDNA kit according to the manufacturer's protocol. cDNA was amplified with Prime Star GXL Polymerase for 14 cycles followed by Blue Pippin size selection (4.5–10 Kb). For capture, 1ug denatured cDNA was used then incubated with Twist Custom Probes at 70 °C for 20 h. Biotynaylated cDNAs were pulled down with streptavidin beads and washed with Twist hybridization buffers to reduce non-specific binding. Targeted cDNA library was amplified 11 cycles with Takara LA Taq yielding 650 ng of enriched cDNA for library prep. SMRTbell Template Prep Kit 1.0 post exonuclease was used for library prep followed by a Blue Pippin size selection (4Kb to 50KB). Post size selection yielded 120 ng of DNA. Sequencing primer version 3.0 was annealed and bound using polymerase version 2.0. The bound complex was cleaned with PB Ampure beads and loaded onto PacBio Sequel instrument by diffusion at 6pM.

**Processing of PacBio raw data**. Iso-Seq software was used for initial post-processing of raw PacBio data. For lrCaptureSeq experiments, reads of insert were generated from PacBio raw reads using ConsensusTools.sh with the parameters–minFullPasses 1 –minPredictedAccuracy 80 –parameters /smrtanalysis/current/analysis/etc/algorithm_parameters/2014-09/. From the reads of insert full-length, non-chimeric reads (FLNC reads) were generated using pbtranscript.py classify with the parameters –min_seq_len 500 and presence of 5′ and 3′ Clontech primers in addition to a polyA tail preceding the 3′ primer. For $Megf11$ PCR product sequencing, parameters were the same except that full-length reads were distinguished by the presence of $Megf11$-specific primer sequences (5′ GGCTCCGGGGTATAGGA; 3′ sequence CTGGCTGCATTGCATTGG for $Megf11$ long or GGTGTCCAATAAAGTC for $Megf11$ short).

**Isoform level clustering**. Clustering of FLNC reads into isoforms was performed using ToFU, which consists of two parts: (1) Isoform-level clustering algorithm ICE (Iterative Clustering for Error Correction), used to generate consensus isoforms; and (2) Quiver, used to polish consensus isoforms. Transcript isoforms were generated using the ToFU_wrap script with the parameters –bin_manual "(0,4,6,9,30)" –quiver –hq_quiver_min_accuracy 0.99 (0.98 for $Megf11$ PCR data). This generated high-quality full-length transcripts with ≥ 99% post correction accuracy (≥ 98% for $Megf11$ PCR data). Isoforms were aligned to the mouse genome mm10 using GMAP (version 1.3.3b) with default values of alignment accuracy (0.85) and coverage (0.99). To prevent over clustering based on 5′ end lengths, redundant clusters were removed by collapsing all transcripts that share exactly the same exon structure. To minimize the impact truncated mRNAs may have on inflating isoform numbers, we set a threshold of ≥ 2 independent full-length reads that must cluster together in order to define an isoform.

To generate the entire isoform catalog, the complete dataset (all timepoints, retina and cortex) was analyzed using the cluster function of Iso-Seq (version 3), with default parameters. Only the highest-quality full-length reads (≥99% accuracy or QV ≥ 20) from each experiment were passed to this analysis. At the conclusion of Iso-Seq 8287 isoforms of our 30 genes were identified. HQ reads were mapped to the genome (mm10 for mouse, hg19 for human) Cupcake ToFU[67] was used to further reduce overclustering of isoform subdivisions.

Finally, additional filtering of putative spurious isoforms was performed with our IsoPops software. The goal of this filtering was to remove artifacts arising from cDNA truncations or poly-A mispriming within genomic DNA. Details of the filtering methodology are provided below in the section describing the software package. Applying these filters yielded the final catalog of 4116 isoforms. We did not exclude isoforms that contained non-canonical junctions, because many such isoforms were highly abundant; however, even if they were excluded, overall isoform counts would be only slightly reduced (Supplementary Fig. 2c).

The final isoform catalog specified not only the number of isoforms, but also the number of full-length reads obtained for each isoform. We have reported these read counts for some of our analyses (e.g., Fig. 2c, e; Fig. 3b, d). These data aid in understanding how the overall expression of a particular gene is distributed across its isoform portfolio. We have avoided making conclusions about the expression level of particular isoforms, unless the PacBio data are supported by independent short-read RNA-seq data (e.g., Fig. 6a–d).

**IsoPops R package**. We developed a package of R software for convenient analysis and viewing of PacBio transcriptome sequence output. The IsoPops R package allows users to perform many of the analyses described in this study on their own long-read data.

The package offers the following features. First, it permits filtering of truncated and spurious isoforms to facilitate downstream analysis. Second, it displays maps of exon usage enabling the user to visually compare how isoforms differ. Third, it generates plots summarizing expression levels of isoforms within an individual gene and across a dataset. These include tree plots (Fig. 2e) and a variant on the Lorenz plot that we have termed a jellyfish plot (Fig. 2c). Fourth, it clusters similar isoforms and displays the data in various dimension-reducing plots such as dendrograms and 3-dimensional PCA plots. Fifth, it provides summary statistics such as the length distribution of a gene's isoforms or the number of exons used in each isoform. Finally, it performs cross-correlations, enabling the user to ask if certain exons tend to appear together in the same transcripts. Methods relevant to these features are described below.

1. Filtering: The IsoPops isoform filtering process consists of three steps: First, transcripts containing fewer than $n$ exons are removed. For our study, $n$ was set to 4, because we did not expect any such short isoforms for the genes in our dataset. To quantify exon number, we did not reference exon annotations, but instead defined the number of non-contiguous genomic segments (or the number of junctions plus one) as the exon count for each isoform. This filtering step removed most spurious transcripts arising from genomic poly-A mispriming, as these sequences typically contained only a single exon as defined by this mapping procedure. Second, the least abundant 5% of isoforms for each gene were filtered out, on the assumption that these extremely low-abundance isoforms might constitute experimental or biological noise.

Finally, we filtered out truncation artifacts. To identify truncated isoforms, we developed an algorithm designed to filter as thoroughly as possible without discarding potentially valuable unique transcripts. In particular, we wanted to preserve all unique splicing events and tolerate unique transcription start sites (TSS) and transcription termination sites (TTS) modestly. The algorithm compares the set of exon boundaries (coordinates of acceptor and donor splice sites) for an isoform pair A and B and applies the following two rules. Rule 1: If all the exon boundaries in B form a contiguous subset of the exon boundaries in A, then B is a truncation of A. We required the subset to be contiguous to avoid filtering transcripts with retained introns. Rule 2: If all three of the following conditions are met, B is a truncation of A. (1) The TSS of B falls within an exon in A; (2) the TTS of B is either found in A or within/beyond the 3′-most exon of the gene; (3) internal exon boundaries of B (i.e., excluding the 5′- and 3′-most exon boundaries of B) are a contiguous subset of A.

2. Pearson correlation: This function enables analysis of exon co-occurrence across isoforms. Each isoform in a given gene was labeled with a series of binary values representing the exons called within its cDNA sequence. Exon calls were determined by searching for exact matches of either the first 30 bp or last 30 bp of each exon within the transcript. Exon definitions were derived from PacBio isofom GFF file. Isoforms were weighted by their full-length read counts before pairwise Pearson correlations between exon calls were calculated.

3. K-mer vectorization: IsoPops enables quantification of sequence differences between isoforms. To quantify relative differences between isoforms, we calculated the Euclidean distances between vectorizations of each isoform's cDNA sequence (or their predicted ORF amino acid sequence). We used the text2vec R package to generate a vector for each isoform, where each element in the vector equals the number of times a certain k-mer (sequence fragment) appears within the isoform. We counted all possible 6-mers within isoforms, choosing $k = 6$ to maximize k-mer count uniqueness between isoforms without requiring excessive computational resources. Each isoform's vector of k-mer counts was then normalized to sum to 1, so that isoform distances calculated from these vectors would not be dominated by differences in length between transcripts.

4. Isoform clustering: To cluster isoforms, we calculated pairwise euclidean distances between isoforms' k-mer count vectorizations. We then performed hierarchical agglomerative clustering using the R base algorithm hclust using default settings and the "complete" agglomeration method. Dendrogram plots of clusterings were generated by the dendextend R package.

5. Dimension reduction: PCA and t-SNE were performed directly on the k-mer count vectorizations. We used the R base algorithm prcomp for PCA with default settings. For t-SNE, we ran the Rtsne package's algorithm for exact t-SNE (theta = 0, maximum iterations = 1000, perplexity = 35), which includes a round of PCA for data pre-processing. t-SNE results are plotted in the same number of dimensions as output by the algorithm (i.e., 3D t-SNE plots were generated with ndim = 3).

6. Lorenz (Jellyfish) plot: Cumulative percent abundance was calculated independently for the isoforms of each gene. First, full-length read counts were normalized across the gene and labeled percent abundance. Next, isoforms for a given gene were rank-ordered by percent abundance in descending order. Finally, a cumulative percent abundance was calculated for each isoform, via partial summation of percent abundances in descending order. Isoforms were then plotted in this order along the y-axis and positioned according to cumulative percent abundance along the x-axis.

**ORF prediction and proteomics reference library**. Sqanti[68] (version 1.2) was used for ORF prediction and genomic correction of PacBio isoforms. To generate the lrCaptureSeq reference peptide library for proteomics, amino acid sequences were trypsinized in silico using the python program *trypsin* with default settings. The proline rule was followed which did not cut lysine or arginine if it immediately preceded a proline.

**RNA-seq analysis**. RNA-seq fastq files were downloaded from NCBI GEO (www.ncbi.nlm.nih.gov/geo/) and the data were mapped with Hisat2 (version 2.1.0) to reference build mm10 (for mouse), hg19 (for human), bosTau8 (bovine), danRer11 (zebrafish), and rn6 (rat). Datasets GSE101986 and GSE74660 were quantified with Cufflinks (version 2.2.1). Datasets GSE94437, GSE101544, GSE59911 and GSE84932 were quantified with StringTie (version 1.3.3b). All reference annotations for isoform quantification analysis were generated from corresponding reference GTF files merged with the Iso-Seq GFF output using the top 3 most abundant isoforms for each of the 30 genes.

**Isoform predictions from RNA-seq data**. Computational prediction of isoforms was performed on the RNA-seq data set GSE101986 and GSE79416 using Cufflinks (version 2.2.1) or Stringtie (version 1.3.3b) without a reference assembly. Resulting assemblies were merged using Cuffmerge to create the final reference assembly. Isoform matching between datasets was performed using Sqanti. Isoforms were

considered a match if they were identified as full-splice match by Sqanti. All other isoforms were considered non-matching.

**Matching of lrCaptureSeq isoforms to other databases**. Sqanti was used for validation of isoforms in public databases, as well as Cufflinks/Stringtie predicted isoform databases. Validation was performed using the reference GTF (either from computational assembly, NCBI RefSeq, or UCSC Genes) as input. Isoforms were validated if they returned a full-splice match to the reference. All other isoforms were considered distinct.

**Validation of isoform splice junctions and 5′ ends**. Junction coverage of PacBio isoforms by RNA-seq data was assessed using Sqanti software. The junction input file for Sqanti was generated using STAR (STAR_2.6.0a) by mapping mouse retina and cortex RNA-seq data (GSE101986 and GSE79416) to the mm10 genome with a custom index made using the PacBio GFF output. Junctions were classified as either canonical (GT-AG, GC-AG, and AT-AC) or noncanonical (all other combinations).

CAGE RNA-seq data (available from the DDJB sequence read archive https://ddbj.nig.ac.jp/) from adult mouse retina (DRA002410) was used for validation of 5′ ends. Samples Sham1 (DRX019832), Sham2 (DRX019833), and Sham3 (DRX019834) were aligned to the genome (mm10) using Hisat2. Read coverage at exon 1 of the lrCaptureSeq isoforms was determined using BedTools (version 2.29.2). CAGE data coverage across normalized isoform lengths was performed using Qualimap (version 2.2.1).

**Chromatin accessibility**. Publicly available ATAC-seq data was used to assess chromatin accessibility (i.e., putative promoter sites) in mouse and human retina[69–71]. DNAse I hypersensitivity data from the ENCODE project was used for assessment of mouse cortex[72]. All raw fastq files were downloaded from SRA or aligned bam files from ENCODE data portal. Reads were trimmed using fastqc (version 0.11.3) and trim galore (version 0.4.1) and mapped to either the mm9 or hg19 genomes using bowtie2 (version 2.2.5). Aligned bam files were filtered for quality (>Q30) and mitochondrial and blacklisted regions were removed. Files were converted to bigwigs using deeptools (version 3.1.0) and visualized in IGV (version 2.4.16). All tracks from the same experiment are group scaled.

**Shannon diversity index**. The Shannon index was calculated with the R package Vegan[73] according to the following equation:

$$H\prime = - \sum p_i \ln p_i \qquad (1)$$

In this equation $_{p_i}$ is the proportion of isoforms found in a gene:

$$p_i = n_i/N \qquad (2)$$

where $n_i$ is the number of reads for isoform $i$ and $N$ is the total number of reads for a gene.

**Sashimi plots**. Sashimi plots were generated using Gviz (version 1.24.0) with the PacBio generated GFF file. The reads for the plot were generated by mapping the PacBio FLNC.fastq (≥ 85% accuracy) file to the genome (mm10, hg19) with GMAP (version 2014-09-30). Because the FLNC reads had relatively high error rates that had not been filtered out like in our final datasets, and because expression varied by gene, minimum junction coverage was variable for each plot. Minimum junction coverage was set to 60 for *Crb1* mouse retina, 4 for *Crb1* Cortex, 11 for human *CRB1*, and 4 for *Megf11*.

**Single-cell RNA-seq**. Raw scRNAseq data profiling mouse retinal development (GSE118614)[50] were aligned to a custom mm10 mouse genome/transcriptome using CellRanger (v3.0, 10X Genomics). mm10 reference genome and transcriptome were downloaded from 10X Genomics and the GTF file was modified to identify the dominant *Crb1* isoforms (*Crb1-A* and *Crb1-B*) as independent genes. As the CellRanger count function only considers alignments that uniquely map to a single gene, output files only report reads that map within the independent 3′ exons or splice into these from the most distal last shared exon.

The resulting CellRanger count output matrices (expression and barcode matrices) were imported into R and manually aggregated. The aggregate matrices were used to generate a Monocle (v3.0) cell data set, using gene annotations as the feature matrix. Transcript expression across cells was normalized to transcript copies per 10,000 (CPT)[74]. Established cell type annotations were imported from GSE118614[50].

**BaseScope in situ hybridization**. Eyes were enucleated and retinas were dissected from the eyecup, washed in PBS, and fixed at RT for 24 h in PBS supplemented with 4% formaldehyde. Retinas were cryoprotected by osmotic equilibrium overnight at 4° in PBS supplemented with 30% sucrose. Retinas were imbedded in Tissue Freezing Medium and flash frozen in 2-methyl butane chilled by dry ice. Retina tangential sections were cut to 18 μm on a Thermo Scientific Microm HM 550 Cryostat and adhered to Superfrost Plus slides.

Probes were designed against splice junctions to detect various splicing events (see Supplementary Table 2 for sequences). Probe detection was performed using the Red detection kit. BaseScope in situ hybridization was performed according to the manufacturers protocol with slight modifications. Fixed frozen retinas were baked in an oven at 60 °C for 1 h then proceeded with standard fixed frozen pretreatment conditions with the following exceptions: Incubation in Pretreatment 2 was reduced to 2 min and Pretreatment 3 was reduced to 13 min at RT. BaseScope probes were added to the tissue and hybridized for 2 h at 40 °C. Slides were washed with wash buffer and probes were detected using the Red Singleplex detection kit. Immunostaining was performed after probe detection by incubation with primary antibodies overnight. For *Megf11* BaseScope, α-Calbindin antibodies were used to label starburst amacrine cells and horizontal cells. Tissue was washed 3 times with PBS and Alexa 488-conjugated secondary antibodies were applied and incubated for 1 h at RT. Slides were washed once again and coverslips mounted.

**Expression of CRB1 isoforms in K562 cells**. Tagged CRB1 constructs were built by cloning YFP in-frame at the C-terminus of CRB1-A and CRB1-B. The tagged constructs were cloned into the pCAG-YFP plasmid (Addgene #11180).

K562 cells (ATCC® CCL-243™) were obtained from, validated by, and mycoplasma tested by ATCC. The cells were cultured in Dulbecco's Modified Eagle's Medium (DMEM) with 10% bovine growth serum, 4.5 g/L D-glucose, 2.0 mM L-glutamine, 1% Penicillin/Streptomycin in 10 cm cell culture dishes. Cells were passaged every 2–3 days before reaching 2 million cells/ml. Cells were transfected using the Amaxa® Cell Line Nucleofector® Kit V following instructions in the K562 nucleofection manual. Specifically, aliquots of 1 million cells were pelleted through centrifuging at $200 \times g$ for 5 min at room temperature in Eppendorf tubes. Supernatant was completely and cell pellets were suspend in 100 ul Nucleofector® solution per sample. A total of 2 ug of plasmid DNA (pCAG:Crb1A-YFP, pCAG:Crb1B-YFP, or pCAG:YFP) were added and gently mixed with the suspended cells. Cell and DNA mixture were transfected into cuvettes, inserted into the Nucleofector® Cuvette Holder, and transfected with program T-016. Cuvettes were taken out of the holder after program is completed and immediately added with 500ul of pre-equilibrated cultured medium. These transfected cells were then divided and transferred into two wells of the 24-well glass bottom dish (MatTek Corporation). Cells were imaged 24-h post transfection with an inverted confocal microscope (Nikon).

**Retina thin sectioning and electron microscopy**. Mice were anesthetized with isoflurane followed by decapitation. Superior retina was marked with a low-temperature cautery to track orientation. Eyes were enucleated and fixed overnight at RT in Glut Buffer (40 mM MOPS, 0.005% CaCl$_2$, 2% formaldehyde, 2% glutaraldehyde in H$_2$O). The dorsal-ventral axis was marked at the time of dissection so that superior and inferior retina could subsequently be identified in thin sections. Eyes were transferred to a fresh tube containing PBS for storage 4 °C until prepped for embedding.

For thin sections, the cornea was removed from the eyecup and the eyecup was immersed in 2% osmium tetroxide in 0.1% cacodylate buffer. The eyecup was then dehydrated and embedded in Epon 812 resin. Semi-thin sections of 0.5 µm were cut through the optic nerve head from superior to inferior retina. The sections were counterstained with 1% methylene blue and imaged on an Olympus IX81 bright-filed microscope.

For electron microscopy, tissue was fixed and embedded as for thin sections. Far peripheral retina was trimmed and 65–75 µm sections were prepared on a Leica EM CU7 ultramicrotome. Sections were prepared separately from superior and inferior hemisections of each retina, and counterstained with a solution of 2% uranyl acetate + 3.5% lead citrate. Imaging was performed on a JEM-1400 electron microscope equipped with an Orius 1000 camera.

**Retina nuclei counting**. Retina semi-thin sections were tile scanned on an Olympus IX81 bright-filed microscope with a 60X oil objective and stitched together with cellSens software. Using Fiji software, a segmented line was drawn from the optic nerve head to the periphery for both superior and inferior retina. At intervals of 500 µm, four boxes of 100 µm were drawn encapsulating the outer nuclear layer so that the center of the box was a factor of 500 µm from the optic nerve head. For each hemisphere of the retina, four boxes were made. Using the count function in ImageJ, the total number of nuclei encapsulated by each box were counted at each position. Counts were averaged across each position and plotted as well as total counts for all 8 measurements for each retina.

**Assessment of OLM junctions by electron microscopy**. Each section, comprising ~90% of one retinal hemisection (far peripheral retina was trimmed during sectioning), was evaluated on the electron microscope for OLM gaps. Each potential gap was imaged and gaps were subsequently confirmed offline by evaluating the presence of electron-dense OLM junctions on the inner segments of imaged photoreceptors. The number of gaps per section was quantified, along with the size of each gap, using Fiji software. For quantification and statistics, wild-type and *null*/+ heterozygous controls were grouped together, since neither genotype showed any OLM gaps.

**Retina serial sectioning with Western blotting**. For the serial sectioning-Western blotting method[52,53], mice were anesthetized with isoflurane followed by decapitation. Eyes were enucleated and dissected in ice-cold Ringer's solution. A retina punch (2 mm diameter) was cut from the eyecup with a surgical trephine positioned adjacent next to the optic disc, transferred onto PVDF membrane with the photoreceptor layer facing up, flat mounted between two glass slides separated by plastic spacers (ca. 240 µm) and frozen on dry ice. The retina surface was aligned with the cutting plane of a cryostat and uneven edges were trimmed away. Progressive 10-µm or 20-µm tangential sections were collected—depending upon endpoint of sectioning (photoreceptors or inner retina, respectively). Blotting was performed with antibodies to CRB1-B, rhodopsin, and GAPDH (see Supplementary Table 2 for antibody details).

**Sample preparation for proteomics**. Juvenile (P14) mice were anesthetized with isoflurane followed by decapitation. Eyes were enucleated and dissected out of the eyecup in Ringers solution (154 mM NaCl, 5.6 mM KCl, 1 mM MgCl$_2$, 2.2 mM CaCl$_2$, 10 mM glucose, 20 mM HEPES). For trypsin release of ectodomains, retinas were placed in 100 µl Ringers solution containing 5 µg trypsin/lys-c. This preparation was incubated at RT for 10 min with periodic gentle mixing. Contents were then centrifuged at $300 \times g$ for 1.5 min and the supernatant was transferred to a new tube. Urea was added to protein mixture to 8 M then incubated at 50 °C. After 1 h incubation, DTT was added to a final concentration of 10 mM and incubated for 15 min at 50 °C. Peptides were alkylated by adding 3.25 µl of 20 mM Iodoacetamide and incubated for 30 min at room temperature in the dark. Reaction was quenched by adding DTT to 50 mM final concentration. Mixture was diluted 1:3 with ~270 µl of ammonium bicarbonate. Mixture was further digested overnight by adding 1 µg of trypsin/lys-c at 37 °C.

For cell-surface biotin labeling of membrane proteins[75], retinas were dissected out of the eyecup into ice-cold HBSS. Retinas were washed with HBSS followed by incubation in HBSS supplemented with EZ-Link Sulfo_NHS-SS-Biotin (0.5 mg/ml in HBSS) for 45 min on ice. Retinas were then washed 3X with HBSS + 100µM lysine to quench remaining reactive esters. Retinas were then collected in 400 µl (200 µl/retina) lysis buffer (1% Triton X-100, 20 mM Tris, 50 mM NaCl, 0.1% SDS, 1 mM EDTA). Retinas were homogenized using short pulses on a sonicator. The lysate was centrifuged at $21,000 \times g$ for 20 min at 4 °C and the soluble fraction was collected. For immunoprecipitation, 75 µg of protein lysate was mixed with 100 µl of Streptavidin Magnetic Beads (Pierce™) and incubated at room temperature while rotating. Streptavidin/biotin complex was sequestered using a magnet and washed with lysis buffer. Proteins were eluted from the beads by incubation with elution buffer (PBS with 0.1% SDS 100 mM DTT) at 50 °C for 30 min. Experimental samples (input, biotin enriched, and non-biotin labeled negative control) were mixed with 4× SDS-PAGE sample buffer and incubated on a heat block at 90 °C for 10 min. Samples were then loaded on a 4–15% mini PROTEAN TGX Stain-Free protein gel. Electrophoresis was carried out at 65 V through the stacking gel then adjusted to 100 V until the dye front reached the end of the gel.

After electrophoresis, the gel was washed twice with H$_2$O, fixed with 50% methanol, 7% acetic acid for 20 min and stained with colloidal Coomassie based GelCode Blue Stain reagent (Thermo Fischer Scientific, cat # 24590) for 30 min. The gel was destained with distilled water at 4 °C for 2 h while rocking. Protein bands were imaged on a Bio-Rad ChemiDoc Touch imager. Using a clean razor blade, bands between 75 and 250 kDa were excised, cut into ~1 × 1 mm pieces and collected in 0.5 ml siliconized (low retention) centrifuge tube. Gel pieces were destained with 200 µl of Destaining Solution (50 mM ammonium bicarbonate, NH$_4$HCO$_3$ in 50:50 acetonitrile:water) at 37 °C for 30 min with shaking. Solution was removed and replaced with 200 µl of Destaining Solution and incubated again at 37 °C for 30 min with shaking. Solution was removed from the gel pieces and peptides were reduced with 20 µl of 20 mM DTT in 50 mM ammonium bicarbonate buffer (pH 7.8)at 60 °C for 15 min. Cysteines were alkylated by adding 50 µl of the alkylation buffer (ammonium bicarbonate buffer with 50 mM Iodoacetamide) and incubated in the dark at room temparature for 1 h. Alkylation buffer was removed from tubes and replaced with 200 µl destaining buffer. Samples were incubated for 30 min at 37 °C with shaking, buffer removed, and washed again with destaining buffer. Gel pieces were dehydrated with 75 µl of acetonitrile and incubated at room temperature for 15 min. Acetonitrile was removed from tubes and shrunken gel pieces were left to dry for 15 min. Trypsin/lys-c (5 ng/µl in 25 µl of ammonium bicarbonate buffer) was added to gel pieces and incubated for 1 h at room temperature. An additional 25 µl of ammonium bicarbonate buffer was added to the tubes and incubated overnight at 37 °C. Sample volume was brought to 125 µl with distilled water, and liquid containing trypsinized peptides was placed in a clean siliconized 0.5 ml tube.

**Mass spectrometry analysis**. A total of 2 µl aliquots of tryptic digests were analyzed by LC-MS/MS using a nanoAcquity UPLC system coupled to a Synapt G2 HDMS mass spectrometer (Waters Corp, Milford, MA). Peptides were initially trapped on a 180 µm × 20 mm Symmetry C18 column (at the 5 µl/min flow rate for 3 min in 99.9% water, 0.1% formic acid). Peptide separation was then performed on a 75 µm × 150 mm column filled with the 1.7 µm C18 BEH resin (Waters) using the 6–30% acetonitrile gradient with 0.1% formic acid for 90 min at the flow rate of 0.3 µl/min at 35 °C. Eluted peptides were sprayed into the ion source of Synapt G2 using the 10 µm PicoTip emitter (Waters) at the voltage of 3.0 kV.

Each sample was subjected to a data-independent analysis (HDMSE) using ion mobility workflow for simultaneous peptide quantitation and identification. For robust peak detection and alignment of individual peptides across all HDMSE runs we performed automatic alignment of ion chromatography peaks representing the same mass/retention time features using Progenesis QI software. To perform peptide assignment to the ion features, ProteinLynx Global Server version 2.5.1 (Waters) was used to generate searchable files that were submitted to the IdentityE search engine incorporated into Progenesis QI for Proteomics (version 4.1). Precursor mass tolerance for the database search was 5 parts per million; fragment mass tolerance was 0.50 Da. For peptide identification we searched against the Iso-Seq/lrCaptureSeq custom database described above (see ORF Prediction subsection), as well as the UniProtKb mouse database. Post-processing using Protein and Peptide Prophet software (Scaffold 4.4) was used to evaluate confidence in peptide matches and to control false discovery rates (FDRs). Spectrum-peptide matches with <50% confidence score were excluded from further analysis. FDR cutoffs for accepting peptide and protein identifications were 1%. In addition, a target-decoy analysis was performed by searching a decoy database—reversed mouse UniProt 2016 database. The FDR measured in this way was 0.16%.

Using these procedures we identified unannotated peptides (Supplementary Data 3) as those that were detected in the custom Iso-Seq database but not in the UniProt database. To distinguish newly discovered peptides from known annotated peptides containing posttranslational modifications, we conducted additional UniProt database searches using the most common protein modifications, including phosphorylation at S,T and Y; glutamylation at E; acetylation at K; methylation at D and E. Each of these modifications was tested individually via a separate database search. No potential false identifications were found. Upon identification of spectra matching unannotated peptides, we took additional quality control steps to ensure the reliability of these matches. First, spectra were inspected manually to confirm alignment of measured and predicted peaks. Second, we compared search engine scores for spectra matching unannotated peptides to those matching known peptides. Both groups had similar search engine scores (Score range: Unannotated = 15–150; Known = 15–130. Mean score: Unannotated = 42.2; Known = 49.9. Median score: Unannotated 38.3; Known = 41.3. Match confidence for median score: Unannotated = 91.8; Known = 91.2).

**Western blotting**. Retinas from littermate WT and *Crb1* mutant mice were briefly sonicated and vortexed in 400 µl of the lysis buffer containing 2% SDS in PBS plus protease inhibitor cocktail (cOmplete; Roche). The lysates were spun at $20,000 \times g$ for 10 min at 22 °C, supernatants collected and total protein concentration determined by the *DC* protein assay kit (Bio-Rad). Using lysis buffer, the volumes were adjusted to normalize the lysates by total protein concentration before adding 4× SDS-PAGE buffer containing 400 mM DTT and heating the lysates for 10 min at 90 °C. Equal volumes of the lysates, each containing 15 µg total protein, were subjected to SDS-PAGE and proteins were transferred to polyvinylidene fluoride membranes (Bio-Rad). The membranes were blocked in the Odyssey blocking buffer (LiCor Bioscience) and incubated with the appropriate primary antibodies (anti-CRB1-B, anti-Phosducin, and anti-ABCA4) and Alexa Fluor 680 or 800 conjugated secondary antibodies (Invitrogen). Protein bands were imaged by the Odyssey CLx infrared imaging system (LiCor Bioscience). See Supplementary Table 2 for additional information on primary and secondary antibodies.

To separate soluble and insoluble proteins, mouse retinas were briefly sonicated and hypotonically shocked in 300 µl of water on ice. The lysed retinal suspensions were spun at $20,000 \times g$ at 4 °C for 20 min, the resulting supernatant was collected and the pellet was rinsed once with water. The pellet and supernatant were reconstituted in a final volume of 400 µL lysis buffer, containing 2% SDS, 1x PBS, and protease inhibitor cocktail (cOmplete; Roche) Equal volume aliquots of these lysates were used as described above for Western blotting.

**Statistics and reproducibility**. The lrCaptureSeq experiments were performed once for each condition/age (four mouse retina timepoints; one mouse brain timepoint; one human retina condition). Mouse experiments used one C57Bl6/J animal for each condition. Human experiments used tissue from a single donor (male, age 59). Each proteomics strategy (i.e., cell surface biotinylation and trypsin ectodomain release) was performed once, although the biotinylation and pull-down conditions were worked out in pilot experiments. The gel shown in Fig. 3f is the same one used for the proteomics experiment and was representative of the pilot experiments using similar conditions. For each proteomics strategy, P14 mouse retinal tissue was pooled from multiple littermates. Even though the large-scale sequencing and mass spectrometry experiments were performed only once, we replicated key results using different experimental approaches, such as corroboration with independent short-read datasets (Fig. 6c, d; Supplementary Fig. 2b); CAGE-seq datasets (Supplementary Fig. 1d); and qPCR (Supplementary Fig. 5b, c).

PacBio sequencing was performed on two different retinal *Megf11* RT-PCR reactions as shown in Supplementary Fig. 4. Together with lrCaptureSeq, therefore, we had 3 independent PacBio datasets for the *Megf11* gene. RT-PCR gel images shown in Supplementary Fig. 4a are representative of many such reactions that were performed. These images are also representative of the reactions that were used for PacBio sequencing.

Mouse mutant phenotypes reported in Figs. 9, 10, and Supplementary Fig. 7b–f have been observed in multiple animals and multiple litters from different founder strains of the mutant alleles. Since the phenotypes were consistent, these groups were pooled for the analysis and reported as a single experiment. Photomicrographs in these figures are representative of the phenotypes, or the range of phenotypes, observed across animals of a given genotype.

Western blots of CRB1-B expression in *delB* mutants (Fig. 7c, d) are representative examples of results obtained from 3 independent experiments (i.e., 3 biological replicates). The CRB1-B blot from *null* mutant (Supplementary Fig. 7b) is a representative example of an experiment that was repeated twice (2 biological replicates). The serial section Western blotting experiment was repeated three times (3 biological replicates); images shown in Fig. 8d are from a single experiment. They are representative examples of the results obtained each time.

Images shown in Figs. 4e and 8c are representative of BaseScope staining that was repeated at least three times on retinal tissue from separate animals. Images of transfected K562 cells (Supplementary Fig. 6c) are representative of two independently transfected tissue culture coverslips, which were imaged in parallel. The *Crb1* qPCR experiment (Supplementary Fig. 5b) was performed once, although results were consistent with smaller pilot experiments in which primers and conditions were being tested. The *Crb1* RT-PCR gel (Supplementary Fig. 5c) was run twice on different RNA samples with identical results.

Statistical analyses comparing two or more groups were performed with Prism software (version 8.3.1). Tests included ordinary one-way ANOVA with Tukey's post-hoc test, and two-way ANOVA with Sidak's post-hoc test. All post-hoc tests were performed with corrections for multiple comparisons.

**Reporting summary**. Further information on research design is available in the Nature Research Reporting Summary linked to this article.

## Data availability
The data that support this study are available from the corresponding author upon reasonable request. Long-read sequencing data generated in this study have been deposited in the NCBI BioProject repository (accession number PRJNA547800). The Supplementary Data 1 and Supplementary Data 2 files specify the sequence, genomic location, and read number for all isoforms within the lrCaptureSeq dataset. Mass spectrometry proteomics data generated in this study have been deposited at the ProteomeXchange Consortium via the PRIDE partner repository with the dataset identifier PXD017290. *Crb1* isoform cDNA sequences described in this study have been deposited at Genbank with the following accession numbers: MT470365 (human *CRB1-A*); MT470366 (human *CRB1-B*); MT470367 (human *CRB1-C*); MT470368 (mouse *Crb1-A*); MT470369 (mouse *Crb1-B*); MT470370 (mouse *Crb1-C*); and MT470371 (mouse *Crb1-A2*).

The source data underlying graphs in Figs. 6c, d, 9j, 10c, d, Supplementary Fig. 5a, b, Supplementary Fig. 6b, and Supplementary Fig. 7c, f are provided in a Source Data file. Also see the Source Data file for full gel images related to Figs. 7c, d, 8d, Supplementary Fig. 5c, and Supplementary Fig. 7b. Source data are provided with this paper.

## Code availability
IsoPops code is available at https://kellycochran.github.io/IsoPops/index.html, licensed under the GNU General Public License v3.0. Source data are provided with this paper.

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

## Acknowledgements

This work was supported by the National Eye Institute (EY024694 and EY030611 to J.N.K., EY026344 to T.A.R., EY5722 to Duke University, EY027844 to B.S.C.); Pew Charitable Trusts (J.N.K.); McKnight Endowment Fund for Neuroscience (J.N.K.); a Duke University School of Medicine Kahn Neurotechnology Award (J.N.K.); Foundation Fighting Blindness (Individual Investigator Award BR-CMM-0619-0767-DUKE to J.N.K.); and Research to Prevent Blindness (Unrestricted Grant to Duke University; Unrestricted Grant to Washington University). Duke Research Computing provided computing resources for data analysis. This project benefitted from data storage capacity provided by the Duke Data Commons, a data storage installation underwritten by the NIH (1S10OD018164-01). We thank the Duke Transgenic and Knockout Mouse Shared Resource for their assistance in making mutant mice, Liz Tseng (PacBio) for assistance with Iso-Seq 3, and Ariane Pereira for mouse colony management.

## Author contributions

Conceptualization: T.A.R., K.C., J.N.K. Methodology: T.A.R., K.C., G.A., M.A.C., W.J.S., P.A.R., B.S.C., A.L., M.-X.H., X.W., E.P., Y.H., A.I., G.H., O.F., N.P.S., V.Y.A., J.N.K. Software: T.A.R., K.C., P.A.R., B.S.C., G.H. Validation: T.A.R., K.C., C.K., J.W., G.A., M.A.C., W.J.S., P.A.R., B.S.C., A.L., M.-X.H., X.W., E.P., G.H., O.F., N.P.S., V.Y.A., J.N.K. Investigation: T.A.R., K.C., C.K., J.W., G.A., M.A.C., W.J.S., P.A.R., B.S.C., Y.H., G.H., O.F., N.P.S., J.N.K. Resources: P.A.R., B.S.C., A.L., M.-X.H. X.W., E.P., A.I., V.Y.A, J.N.K. Data Curation: T.A.R., K.C., P.A.R., B.S.C. Writing—Original Draft: T.A.R., J.N.K. Writing—Review & Editing: T.A.R., K.C., C.K., J.W., G.A., M.A.C., W.J.S., P.A.R., B.S.C., A.L., M.-X.H., X.W., E.P., A.I., G.H., O.F., N.P.S., V.Y.A., J.N.K. Visualization: T.A.R., K.C., P.A.R., B.S.C., J.N.K. Supervision: A.I., O.F., N.P.S., V.Y.A., J.N.K. Funding Acquisition: T.A.R., J.N.K.

## Competing interests

T.A.R. and J.N.K.: Patent pending, "Compositions and Methods for the Diagnosis and Treatment of Retinopathies." A.L., M.-X.H., X.W., and E.P. were employed by Advanced Cell Diagnostics at the time of this project. All other authors declare no competing interests.
