## [Peer Review File · Nature Communications]

Reviewers' comments:

Reviewer #1 (Remarks to the Author):

This is a very well written manuscript describing a comprehensive set of studies that identified an important Crb1 isoform that was not previously known. The manuscript is quite long however, and would benefit by some re-focussing. At 85 pages overall and some 20+ pages of text, it would be improved by moving some of the content to the supplemental data section. I as a reviewer appreciate the concept that the discovery of the new, clearly important disease causing isoform of Crb1 was the result of an unbiased screen - not looking for retina disease genes, nor retina specific genes, but as a result of a screen for neuronal cell surface molecules. However I believe that a major portion of the interest will come from retina disease research community, and moving the retinal disease component to a more prominent position in the abstract and perhaps the title as well will improve the focus. As it now reads, it begins with a strong focus on the screening techniques, and the validation of the identification of the important crb1 disease causing isoform by the KO mouse model is not apparent until the reader is well into it. Overall, the finding does turn on its head over a decade of work on the "other" isoform of Crb1 and many papers on knocking out and replacing this other isoform with mixed results, and this is a major finding of the work, which should be emphasized early in the text.

Reviewer #2 (Remarks to the Author):

This manuscript from Ray et al. addresses a glaring gap in our existing gene annotation, namely how poorly the current annotation catalogs the set of full-length mRNA transcript isoforms. This gap creates a whole lot of confusion and wasted time, as cogently laid out in the manuscript's introduction. The gap reflects a glaring technological blind spot that the authors partially address by developing a capture approach that enables full-length sequencing of mRNAs on the PacBio platform. While this approach is not, by a long shot, a genome-wide approach (30 genes only here), it is very well executed. Moreover, the authors go to surprising lengths to demonstrate the full power of the information they are generating, including an in-depth study of a new CRB1 isoform, featuring an antibody and knockouts that enable the new isoform to be detected and studied functionally in the retina. They show that the newly identified isoform is important for the integrity of junctions in the outer limiting membrane.

This manuscript is a vivid reminder of the limitations of our existing annotation, and it goes all the long way from discovery of thousands of new isoforms to demonstration of the functional importance of one of them. Although I have never before recommended publication of a manuscript without any suggested revisions, in this case it seems most appropriate to do so.

Other comments:

This manuscript is not just refreshingly well-written, it is one of the more lucid manuscripts I have ever read.

The treeplot (e.g., Figure 2D) is a fantastic way to show frequency of transcript isoforms -- I wish this had been used in some of the early RNA-seq studies on alternative splicing, where it was so hard to understand the significance of the diversity that was reported in the absence of this kind of transparency.

Reviewer #3 (Remarks to the Author):

In the ms by Ray et.al, the authors developed a new strategy to characterize the full-length sequence of various mRNA isoforms derived from a target set of genes. The so-termed long-read capture sequencing (lrCaptureSeq) used biotinylated probes to enrich cDNAs from target transcripts, which were then sequenced by using PacBio long read technology. They applied it on 30 genes encoding cell-surface proteins expressed across development in mouse retina and brain, and identified a total of 4116 different isoforms, with a median 50 isoforms per gene. This dataset largely expanded the current isoform annotation and a major fraction of newly discovered isoforms could encode novel protein isoforms, many of which could be supported with MS-based proteomics data. Finally, they focused on one gene CRB1, for which they discovered a novel isoform expressed in photoreceptors. By isoform-specific genetic perturbation, the authors demonstrated the important function of this novel isoforms and explained the previous inconsistent phenotypes between different mouse mutants affecting different isoforms. Overall, by presenting a new strategy (although not the first one) of applications to many different systems, the manuscript should be of interest to broad audience. However, a few points listed below should be addressed before accepted for publication.

- 1) How can the authors be sure of the completeness of their isoform sequences, particularly the 5' end (strand switch is not necessarily happening only at the intact 5' end)? They should use independent approach, such as CAGE and 3' end sequencing to validate both ends of their isoforms.
- 2) As I understood, the authors estimated the cellular abundance of different isoforms based on the number of corresponding pacbio reads. Given the potential biases introduced during capture and pacbio sequencing, such estimation might be inaccurate and this should be addressed.
- 3) The authors selected the cDNA size between 4.5 and 10kb for library preparation. Will this introduce false negatives, i.e. how can they assure no. isoform beyond the size range? They should collect 5'/3' end RNA-sequencing data (see 1)) to estimate false negatives.
- 4) In the part of MS-based proteomics validation, out of 686 peptides, only 35 were not matched to Uniprot, but to their references. This corresponded to a 5% gain, instead of 25% as they claimed (Page 12, line 252). The authors should state clearly what fraction of novel protein isoform as well as annotated isoforms were supported by peptide findings and what are the factors affecting it, e.g. the isoform abundance.
- 5) IN page 19, the authors suggested that the isoform diversity they described is likely to be typical for the upper 10-15% of the isoform diversity distribution. In addition to the fact that 30 genes studied in this ms is of specific function (the authors discussed this point), the neuronal tissues used here is also

special. Likely, the different isoforms identified here are present in different cell types and brain, which consists of more diversified types of cells, would potentially manifest higher isoform diversity. Moreover, when discussing the diversity, not only the number of isoforms, but also the evenness of their abundance distribution are important metrics. Therefore, they should be cautious to generalize their speculation and use also Shannon index as the measurement of isoform diversity.

6) In page 18, the authors suggested that their full-length mRNA data could be integrated with single-cell RNA sequencing (scRNA-seq) data to study the temporal and spatial isoform-specific expression pattern. Given the majority of current scRNA-seq technologies could only capture 3' or 5' end sequences. The potential usage of such integration would be rather limited to those with distinct 5'/3' end, as the case for CRB1 gene.

Reviewer 1

1.1 This is a very well written manuscript describing a comprehensive set of studies that identified an important Crb1 isoform that was not previously known. ... Overall, the finding does turn on its head over a decade of work on the " other' isoform of Crb1 and many papers on knocking out and replacing this other isoform with mixed results.

We thank the reviewer for their kind words and we are glad they appreciate the significance of the CRB1 work.

1.2 The manuscript is quite long however, and would benefit by some re-focussing. At 85 pages overall and some 20+ pages of text, it would be improved by moving some of the content to the supplemental data section.

We agree that the original submission was too long and so we have undertaken significant edits to the Introduction, Results, and Discussion. There are too many to list all of the changes here, but we would like to highlight some of the key changes:

- Details about how the PacBio dataset was generated and clustered have been moved from Results to the Methods.
- The t-SNE plot has been moved from Figure 2 to Supplementary Figure 2.
- Most sections in the Discussion have been trimmed or abbreviated.

Other edits can be noted in the Track Changes version of the Word file.

1.3 I as a reviewer appreciate the concept that the discovery of the new, clearly important disease causing isoform of Crb1 was the result of an unbiased screen - not looking for retina disease genes, nor retina specific genes, but as a result of a screen for neuronal cell surface molecules. However I believe that a major portion of the interest will come from retina disease research community, and moving the retinal disease component to a more prominent position in the abstract and perhaps the title as well will improve the focus. As it now reads, it begins with a strong focus on the screening techniques, and the validation of the identification of the important crb1 disease causing isoform by the KO mouse model is not apparent until the reader is well into it... this is a major finding of the work, which should be emphasized early in the text.

We agree that the two-part structure of the paper is an important component of the story. It is critical for the reader to understand that we did not set out to study *Crb1*, but instead found *Crb1-B* in an unbiased manner. However we appreciate the reviewer's concern that the importance of the *Crb1* part might not be emphasized early enough in the manuscript. To address this concern, we have modified the title and abstract to further highlight the disease/CRB1 part of the story. The efforts at shortening the manuscript (see point 1.2 above) should also help in this direction.

In addition we made numerous changes aimed at setting reader expectations early in the manuscript, so the reader knows from the start that the paper will have two parts. We believe these changes will be helpful for readers who are most interested in the second part of the story. Key changes include:

- Edits to the Introduction aimed at setting the reader's expectations for the two-part narrative structure (e.g. lines 66-68).
- We made sure to end the Introduction by emphasizing the key CRB1 results and their importance (lines 80-84).
- We added a sentence about *Crb1* early in the Results section (lines 109-11) to continue reminding the reader that the paper will soon turn in this direction.

We were reluctant to make more dramatic changes to the overall structure of the paper, since the other two reviewers were very positive about the first half of the paper. We therefore did not want to make changes that would diminish their enthusiasm for this part of the story.

Reviewer #1 (Remarks to the Author):

This is a very well written manuscript describing a comprehensive set of studies that identified an important Crb1 isoform that was not previously known [1.1]. The manuscript is quite long however, and would benefit by some re-focussing. At 85 pages overall and some 20+ pages of text, it would be improved by moving some of the content to the supplemental data section [1.2]. I as a reviewer appreciate the concept that the discovery of the new, clearly important disease causing isoform of Crb1 was the result of an unbiased screen - not looking for retina disease genes, nor retina specific genes, but as a result of a screen for neuronal cell surface molecules. However I believe that a major portion of the interest will come from retina disease research community, and moving the retinal disease component to a more prominent position in the abstract and perhaps the title as well will improve the focus. As it now reads, it begins with a strong focus on the screening techniques, and the validation of the identification of the important crb1 disease causing isoform by the KO mouse model is not apparent until the reader is well into it [1.3]. Overall, the finding does turn on its head over a decade of work on the "other" isoform of Crb1 and many papers on knocking out and replacing this other isoform with mixed results [1.1], and this is a major finding of the work, which should be emphasized early in the text [1.3].

Reviewer 2

This manuscript is a vivid reminder of the limitations of our existing annotation, and it goes all the long way from discovery of thousands of new isoforms to demonstration of the functional importance of one of them. Although I have never before recommended publication of a manuscript without any suggested revisions, in this case it seems most appropriate to do so.... This manuscript is not just refreshingly well-written, it is one of the more lucid manuscripts I have ever read.

We are grateful to the reviewer for these exceedingly kind words.

Reviewer #2 (Remarks to the Author):

This manuscript from Ray et al. addresses a glaring gap in our existing gene annotation, namely how poorly the current annotation catalogs the set of full-length mRNA transcript isoforms. This gap creates a whole lot of confusion and wasted time, as cogently laid out in the manuscript's introduction. The gap reflects a glaring technological blind spot that the authors partially address by developing a capture approach that enables full-length sequencing of mRNAs on the PacBio platform. While this approach is not, by a long shot, a genome-wide approach (30 genes only here), it is very well executed. Moreover, the authors go to surprising lengths to demonstrate the full power of the information they are generating, including an in-depth study of a new CRB1 isoform, featuring an antibody and knockouts that enable the new isoform to be detected and studied functionally in the retina. They show that the newly identified isoform is important for the integrity of junctions in the outer limiting membrane.

This manuscript is a vivid reminder of the limitations of our existing annotation, and it goes all the long way from discovery of thousands of new isoforms to demonstration of the functional importance of one of them. Although I have never before recommended publication of a manuscript without any suggested revisions, in this case it seems most appropriate to do so [2.1].

Other comments:

This manuscript is not just refreshingly well-written, it is one of the more lucid manuscripts I have ever read [2.1].

The treeplot (e.g., Figure 2D) is a fantastic way to show frequency of transcript isoforms -- I wish this had been used in some of the early RNA-seq studies on alternative splicing, where it was so hard to understand the significance of the diversity that was reported in the absence of this kind of transparency.

Reviewer 3

3.1 the manuscript should be of interest to broad audience.

We thank the reviewer for noting the broad appeal of our work

3.2 1) How can the authors be sure of the completeness of their isoform sequences, particularly the 5' end (strand switch is not necessarily happening only at the intact 5' end)? They should use independent approach, such as CAGE and 3' end sequencing to validate both ends of their isoforms.

We thank the reviewer for this suggestion. It is important to validate in particular the 5' end of the transcript, which is always the most uncertain portion of the sequence when libraries are prepared using oligo-dT. We obtained a retinal CAGE-seq dataset (Yasuda et al., 2014; PMID 25407019) and used the data to assess accuracy of the lrCaptureSeq 5' end sequences. The CAGE data were from adult mouse retina, so we validated the 5' ends within the adult mouse retina lrCaptureSeq isoform database (n = 1078 isoforms). Mapping the location of the CAGE reads revealed that 97.5% (1051/1078) of lrCaptureSeq 5' ends had CAGE-seq support. Few if any CAGE reads mapped to other portions of the transcript, further indicating the accuracy of our 5' end identification.

These data are now reported in the Results (lines 131-138) and in Supplementary Fig. 1D,E. Panel D shows CAGE-seq coverage for lrCaptureSeq first exons, while panel E shows CAGE-seq coverage across the entire lrCaptureSeq transcript.

We were not able to identify a suitable 3' end sequencing dataset with which to perform a similar validation. However, we would argue that this is unnecessary, given the following two features of our workflow: 1) we used oligo-dT as the primer for constructing our sequencing libraries; and 2) we validated the vast majority of splice junctions and 5' ends (see Results, lines 131-147). If oligo-dT misprimed from genomic DNA, or otherwise generated a spurious start site for reverse transcription, the resulting isoform sequence would fail to validate at one of the other two stages. Indeed, during our processing pipeline we identified many instances of apparent oligo-dT mispriming at genomic locations. The resulting PacBio reads lacked any splice sites (since they

were reading directly from genomic DNA) and were easily filtered out during our data processing pipeline (see Methods, lines 631-637).

Our confidence in the accuracy of the 3' end sequences is supported by previous PacBio work (Anvar et al., 2018 PMID: 29598823), in which the authors also used the SMARTer cDNA kit for library preparation. In this study the authors found a high degree of cross-validation between short-read 3' end sequencing and their long-read 3' ends.

3.3 2) As I understood, the authors estimated the cellular abundance of different isoforms based on the number of corresponding pacbio reads. Given the potential biases introduced during capture and pacbio sequencing, such estimation might be inaccurate and this should be addressed.

The reviewer raises an important point – especially because there are not yet any broadly accepted methods for quantifying transcript expression levels using PacBio sequencing. Due to this concern, we were very careful not to draw firm conclusions about the expression levels of specific isoforms unless there was independent confirmation in short-read data – for example in the *Crb1* portion of the manuscript (Fig. 5).

We did use PacBio read-count-per-isoform to draw conclusions about how overall expression of a given gene is distributed across isoforms (e.g. Fig. 2C,E; Fig. 3B,D). However, in describing these figures we only drew global conclusions, avoiding making any statements about the expression level of specific isoforms. If the capture and PacBio sequencing processes generated biases that skewed detection of particular isoforms, this is unlikely to affect these global conclusions about the number of isoforms that contribute to overall gene expression.

The reviewer is right that these important considerations should have been discussed in the original submission. This oversight is now corrected via an addition to the Methods section (lines 610-615).

3.4 3) The authors selected the cDNA size between 4.5 and 10kb for library preparation. Will this introduce false negatives, i.e. how can they assure no. isoform beyond the size range?

Based on the reviewer's comment we now realize that the term "size selection" may be confusing (though unfortunately it is the standard term in the field and we need to use it anyway). It is important to emphasize that the size selection produces an enrichment for transcripts of the targeted size but it is by no means a strict cut-off. Indeed one can appreciate in Fig. S2E that we identified many isoforms smaller than 4.5 kb – including *Crb1-B*, which is only 3.0 kb. Nevertheless, because we have depleted smaller transcripts from the sample, the reviewer is entirely correct to point out that we may have missed some smaller transcripts. We now acknowledge this point in the Discussion (lines 385-391).

It is highly unlikely that we missed any transcripts larger than 10 kb, because we intentionally chose genes for which the maximum predicted size (i.e. a transcript including all known exons)

was ≤ 8 kb. In order to exceed 10 kb, there would need to be 2 kb or more of unannotated exon sequence – quite a lot. We cannot exclude that it might have happened, but it is far less of a concern than exclusion of short transcripts, which is why we focused on that possibility in our revised Discussion (lines 385-391).

We have now clarified that we targeted genes with a maximum predicted size of 8 kb – please see Results, line 109 (previously in this line we had simply reported the range of the size selection window).

3.5 They should collect 5'/3' end RNA-sequencing data (see 1)) to estimate false negatives.

It would be great to estimate false negatives, and indeed we attempted to get this to work using the CAGE-seq dataset (see point 3.2 above). But unfortunately we found that it was impossible to estimate false negatives in this way. While we were able to identify “orphan” CAGE reads that did not align to one of our isoforms, it was unclear whether these represented true false negatives. There were two main problems:

1) The gene-of-origin for these orphan reads was often ambiguous. That is to say, the reads often mapped to a genomic region where it might have originated from one of the 30 genes in our study, or it might just as easily have originated from the adjacent gene. We did not attempt a similar analysis with 3' end short-read data, but we anticipate that we would have faced similar challenges in determining the provenance of orphan 3' reads.

2) Enhancer RNAs (eRNAs) and other non-coding RNAs are also detected by the CAGE-seq method (Adiconis et al., 2018 PMC6075671). Therefore, even when the orphan CAGE reads mapped within one of our 30 gene loci, it was still not clear whether it represented a protein coding transcript (i.e. a bona-fide false-negative), or rather an eRNA transcript that was rightly excluded from our PacBio dataset.

Given the difficulty of accurately measuring false negatives, we believe the most appropriate way of addressing this reviewer concern is simply to acknowledge that our isoform catalogs may not be fully complete. This is what we have done in our revised Discussion (lines 385-391)

3.6 In the part of MS-based proteomics validation, out of 686 peptides, only 35 were not matched to Uniprot, but to their references. This corresponded to a 5% gain, instead of 25% as they claimed (Page 12, line 252).

We regret the confusion – the 25% gain referenced in the original manuscript was referring to the comparison of predicted peptides within the Uniprot database vs. the lrCaptureSeq dataset (see graph in Supplementary Fig. 3E). It was not intended to refer to the actual experimental result. This has been clarified in the revised manuscript (lines 244-247).

3.7 The authors should state clearly what fraction of novel protein isoform as well as annotated isoforms were supported by peptide findings and what are the factors affecting it, e.g. the isoform abundance.

On lines 248-250 we state the number of total peptides found by mass spec (686 corresponding to our 30 genes) and the number of peptides that exist only in our IrCaptureSeq predicted peptide database (35). The reviewer wants to know the fraction of annotated isoforms and novel isoforms that are supported by these peptide findings. This is a difficult question to answer, since the peptides are small relative to an entire isoform and usually lie within domains that are shared by multiple isoforms. When one of these shared peptides is detected, it is impossible to know which particular isoform(s) have been supported by the peptide findings. Thus, it is not possible to perform the calculation requested by the reviewer.

If the peptide happens to lie within a region of the protein that is specific to one single isoform, then the interpretation is clearer. But this was quite rare. Most of the peptides we detected are not unique to a single isoform – even the novel peptides may come from domains that are shared across multiple new isoforms. Moreover, the likelihood of finding such a peptide is strongly affected by the amount of isoform-specific protein sequence that exists across the isoforms of a particular gene. This creates a major confound that would likely lead to a significant underestimate of supported isoforms if we were to attempt the calculation suggested by the reviewer using unique peptides alone.

Given these difficulties we think it is most illuminating to consider the data at the peptide level, not at the isoform level. That is why the data on lines 248-250 refer to the novelty of peptides, not isoforms.

3.8 IN page 19, the authors suggested that the isoform diversity they described is likely to be typical for the upper 10-15% of the isoform diversity distribution. In addition to the fact that 30 genes studied in this ms is of specific function (the authors discussed this point), the neuronal tissues used here is also special. Likely, the different isoforms identified here are present in different cell types and brain, which consists of more diversified types of cells, would potentially manifest higher isoform diversity. Moreover, when discussing the diversity, not only the number of isoforms, but also the evenness of their abundance distribution are important metrics. Therefore, they should be cautious to generalize their speculation and use also Shannon index as the measurement of isoform diversity.

The reviewer makes several good points here, with which we agree. Therefore we have removed this speculative assertion from the Discussion. We now simply state that the overall diversity of isoforms across genes and tissues remains to be determined (lines 415-417).

3.9 In page 18, the authors suggested that their full-length mRNA data could be integrated with single-cell RNA sequencing (scRNA-seq) data to study the temporal and spatial isoform-specific expression pattern. Given the majority of current scRNA-seq technologies could only capture 3' or 5' end sequences. The potential usage of such integration would be rather limited to those with distinct 5'/3' end, as the case for CRB1 gene.

The reviewer is absolutely correct about this, and we agree that it merits explicit discussion. We have made an addition to the Discussion addressing this point. We now state that, at present, we

are limited to examining isoforms that differ at the 3' end, although this may change in the future (lines 398-402).

Reviewer #3 (Remarks to the Author):

*In the ms by Ray et.al, the authors developed a new strategy to characterize the full-length sequence of various mRNA isoforms derived from a target set of genes. The so-termed long-read capture sequencing (lrCaptureSeq) used biotinylated probes to enrich cDNAs from target transcripts, which were then sequenced by using PacBio long read technology. They applied it on 30 genes encoding cell-surface proteins expressed across development in mouse retina and brain, and identified a total of 4116 different isoforms, with a median 50 isoforms per gene. This dataset largely expanded the current isoform annotation and a major fraction of newly discovered isoforms could encode novel protein isoforms, many of which could be supported with MS-based proteomics data. Finally, they focused on one gene CRB1, for which they discovered a novel isoform expressed in photoreceptors. By isoform-specific genetic perturbation, the authors demonstrated the important function of this novel isoforms and explained the previous inconsistent phenotypes between different mouse mutants affecting different isoforms. Overall, by presenting a new strategy (although not the first one) of applications to many different systems, **the manuscript should be of interest to broad audience [3.1]**. However, a few points listed below should be addressed before accepted for publication.*

1) How can the authors be sure of the completeness of their isoform sequences, particularly the 5' end (strand switch is not necessarily happening only at the intact 5' end)? They should use independent approach, such as CAGE and 3' end sequencing to validate both ends of their isoforms [3.2].

2) As I understood, the authors estimated the cellular abundance of different isoforms based on the number of corresponding pacbio reads. Given the potential biases introduced during capture and pacbio sequencing, such estimation might be inaccurate and this should be addressed [3.3]

3) The authors selected the cDNA size between 4.5 and 10kb for library preparation. Will this introduce false negatives, i.e. how can they assure no. isoform beyond the size range? [3.4] They should collect 5'/3' end RNA-sequencing data (see 1)) to estimate false negatives [3.5].

4) In the part of MS-based proteomics validation, out of 686 peptides, only 35 were not matched to Uniprot, but to their references. This corresponded to a 5% gain, instead of 25% as they claimed (Page 12, line 252) [3.6]. The authors should state clearly what fraction of novel protein isoform as well as annotated isoforms were supported by peptide findings and what are the factors affecting it, e.g. the isoform abundance [3.7].

5) IN page 19, the authors suggested that the isoform diversity they described is likely to be typical for the upper 10-15% of the isoform diversity distribution. In addition to the fact that 30 genes studied in this ms is of specific function (the authors discussed this point), the neuronal tissues used here is also special. Likely, the different isoforms identified here are present in different cell types and brain, which consists of more diversified types of cells, would potentially manifest higher isoform diversity. Moreover, when discussing the diversity, not only the number

of isoforms, but also the evenness of their abundance distribution are important metrics. Therefore, they should be cautious to generalize their speculation and use also Shannon index as the measurement of isoform diversity [3.8].

6) In page 18, the authors suggested that their full-length mRNA data could be integrated with single-cell RNA sequencing (scRNA-seq) data to study the temporal and spatial isoform-specific expression pattern. Given the majority of current scRNA-seq technologies could only capture 3' or 5' end sequences. The potential usage of such integration would be rather limited to those with distinct 5'/3' end, as the case for CRB1 gene [3.9].

Reviewers' comments:

Reviewer #3 (Remarks to the Author):

All my concerns have been adequately addressed. The manuscript is ready for publication.

Reviewer #4 (Remarks to the Author):

The manuscript submitted by Ray et al. describes a strategy to identify full-length isoforms encoded by individual genes. They show the applicability of their strategy by revealing novel unannotated isoforms expressed in retina and brain. One of their key findings is the discovery of a novel CRB1 isoform, which was previously overlooked. A mutant mouse model subsequently suggested that this isoform is related to accelerated photoreceptor death.

With regard to the validation by mass spectrometry:

1) It is not clear to me how big the protein database is which was used for the discovery of novel peptides and whether it contained all potentially present proteins. Despite the enrichment for cell surface proteins, the samples could still be contaminated with other (e.g. intra-cellular) proteins. In order to avoid matching fragmentation spectra to known peptides from canonical proteins (in contrast to modified peptides from the same database), the authors should redo their analysis using the Sqanti-generated fasta + UniProtKB mouse fasta.

The authors have tried to do this by searching their data with 7 common protein modifications activated. However, it is not clear to me which database was used for this search (Sqanti or full UniProt). In either case, allowing 7 variable modifications increases the search space quite drastically and increases the odds of random matches, subsequently given true positive hits less chance of surviving the FDR cutoff. It is not entirely unexpected that "no potential false identifications were found" by that. Were there any alternative matches to spectra being identified as novel peptides with higher search engine scores in comparison to the unmodified search?

2) The authors have chosen a relatively high peptide level FDR of 5%. How many novel peptides would have survived at an FDR cutoff of 1%? How does the score distribution of novel peptides compare to known peptides? Do they follow the same trend? In the same direction, what is the spectral quality (e.g. S/N, number of peaks) of the spectra identified to be novel peptides vs known peptides?

3) Despite the theoretical validation and given the relatively low number of peptides/spectra to investigate, have the authors considered doing a validation by a) synthetic peptides, b) predicted spectra or c) manual identification? To allow manual investigation, the authors could use tools such as xiSPEC to share annotated spectra of the novel peptides.

Response to reviewers:

4.1) *Reviewer 4: The manuscript submitted by Ray et al. describes a strategy to identify full-length isoforms encoded by individual genes. They show the applicability of their strategy by revealing novel unannotated isoforms expressed in retina and brain. One of their key findings is the discovery of a novel CRB1 isoform, which was previously overlooked. A mutant mouse model subsequently suggested that this isoform is related to accelerated photoreceptor death. With regard to the validation by mass spectrometry:*

We thank Reviewer 4 for their helpful and constructive comments. Our detailed responses to each of Reviewer 4's points are given below. Overall, the reviewer's comments made it clear to us that many additional details needed to be added to the Methods section. For convenience and easy reference, here we summarize the key changes made to the Methods ("Mass spectrometry analysis of retinal samples" subsection):

- We clarify that database searches were performed against both the IrcaptureSeq (i.e. Sqanti) and the UniProt databases. Spectra were assigned to the peptide with the highest search engine score across both databases.
- We provide additional technical details about our database searching strategy and false discovery rate (FDR) analysis:
 - Precursor mass tolerance of the database search was 5 PPM
 - Fragment mass tolerance of the database search was 0.50 Da
 - Peptides below 50% match confidence (PeptideProphet) were excluded from the analysis.
 - FDR cutoffs were 1% for both peptide- and protein-level analysis.
 - Decoy database FDR analysis yielded a measured FDR of 0.16%.
- We clarify how the protein modification searches were performed to prevent an excess of multiple comparisons that might have interfered with detection of a match.
- We highlight steps that were taken to ensure reliability of novel peptide matches, including manual inspection of spectra as well as the comparison of known and novel search engine scores as suggested by the reviewer.

4.2) 1) *It is not clear to me how big the protein database is which was used for the discovery of novel peptides and whether it contained all potentially present proteins. Despite the enrichment for cell surface proteins, the samples could still be contaminated with other (e.g. intra-cellular) proteins. In order avoid matching fragmentation spectra to known peptides from canonical proteins (in contrast to modified peptides from the same database), the authors should redo their analysis using the Sqanti-generated fasta + UniProtKB mouse fasta.*

We regret the lack of clarity: All spectra were indeed searched against the full UniProt mouse database, as well as the Sqanti-generated protein database. The Methods have been modified to clarify this point.

The reviewer is concerned that spectra matching novel peptides might have an equally plausible or better match within the UniProt peptide database. It is highly unlikely that a better UniProt match exists, because we did run our search against both Sqanti and UniProt databases, and we accepted as the matching peptide only the one with the best search engine score across

both databases. (This is the standard operating mode of the Scaffold software that was used for this analysis.)

In response to the reviewer's concern, we have now performed an additional analysis to find out whether spectra assigned to novel peptides might match the UniProt database. We re-searched all of the spectra matching peptides listed in Supplementary Table 3 against the UniProt database (for details on search parameters see #4.1 above, and revised Methods section). No matches were identified. This result supports the notion that these spectra correspond to novel peptides that do not exist in the UniProt database. We have revised the Methods section to emphasize that these spectra did not have matches in UniProt.

4.3) *The authors have tried to do this by searching their data with 7 common protein modifications activated. However, it is not clear to me which database was used for this search (Sqanti or full UniProt). In either case, allowing 7 variable modifications increases the search space quite drastically and increases the odds of random matches, subsequently given true positive hits less chance of surviving the FDR cutoff. It is not entirely unexpected that "no potential false identifications were found" by that.*

Again, we regret the lack of clarity in the initial Methods section. We used the UniProt database for these searches, to rule out the possibility that modified UniProt peptides were misidentified as novel peptides. We did not perform the searches for all 7 modifications at the same time, but instead did them one at a time. This strategy prevents overabundance of multiple comparisons that would lead to an impossibly high bar for FDR cutoff. In the revised Methods section we now make these points more clearly.

4.4) *Were there any alternative matches to spectra being identified as novel peptides with higher search engine scores in comparison to the unmodified search?*

As noted in point 4.2 above, Scaffold software assigns each spectrum to a single peptide that had the highest search engine score. Therefore, for any spectra assigned to a novel peptide (i.e. a peptide found in the Sqanti database but not in UniProt), any other matches must have had lower search engine scores. This includes the searches for modified peptides.

Therefore, even if an "alternate match" within the UniProt database increased its score between the unmodified and modified conditions, the increase was still insufficient to make the score higher than the match to the Sqanti novel peptide. For this reason we do not see the point in conducting any in-depth analysis of those alternate matches, as this is unlikely to change our conclusions.

4.5) 2) *The authors have chosen a relatively high peptide level FDR of 5%. How many novel peptides would have survived at an FDR cutoff of 1%?*

We thank the reviewer for catching an error in our previous submission: We actually used a 1% peptide FDR cutoff in our initial analysis. In response to the reviewer's question, we redid our analysis so as to compare the effects of using a 5% vs 1% FDR cutoff. Across both databases (Sqanti & UniProt), 2398 peptides were identified at the 5% cutoff, versus 2397 at 1% cutoff. The one peptide that dropped out was not a novel peptide. Protein identifications did not change. We have revised the Methods section to make clear that both peptide- and protein-level FDRs are 1%.

In addition, to address the reviewer's broader concern about the statistical rigor of the novel peptide assignments, we now provide further details of the target-decoy FDR analysis that was mentioned in the previous submission. By searching a decoy database (reversed peptides from both the UniProt and the Sqanti isoform databases), we found that the false discovery rate was 0.16%. Therefore, we have high statistical confidence in the assignments of spectra to novel peptides.

4.6) *How does the score distribution of novel peptides compare to known peptides? Do they follow the same trend? In the same direction, what is the spectral quality (e.g. S/N, number of peaks) of the spectra identified to be novel peptides vs known peptides?*

Search engine scores for known and novel peptides were similar:

	Novel peptides	Known peptides
Score range	15-150	15-130
Mean score	42.2	49.9
Median score	38.3	41.3
Match confidence for median score (PeptideProphet)	91.8%	91.2%

These data demonstrate that the novel peptides were not particularly weak hits, but instead were representative of the entire distribution of peptide-spectrum matches. We now note this important fact in the revised Methods.

As for the spectral quality: Our software did not allow us to directly assess signal-to-noise or number of peaks in a large-scale manner, but manual inspection of the spectra representing novel peptides did not reveal any obvious quality issues – they were similar to the other spectra in our dataset. Additionally, since poor quality spectra will generate lower search engine scores, the score data noted above supports the notion that the spectra matching novel peptides were not especially low in quality.

Both of these points raised by the reviewer go towards the critical issue of the accuracy of our peptide assignments: If the spectra matching novel peptides are low quality, this could reduce the search engine score for the true correct match and raise the chances that an incorrect match will garner the highest search engine score by chance. To address this concern, we now clarify in the revised Methods section several methodological aspects that pertain to the accuracy of peptide assignments. First, we now describe the quality control steps that were taken during our analysis. This included discarding spectra from the analysis if their match was below a specified confidence level (as measured by PeptideProphet software), as well as manual inspection of spectra peak alignments. Second, we now specify the mass tolerance of the database search (see section 4.1 above), which was quite stringent. The requirement for high mass accuracy reduces the chances of false matches. Finally, it is important to note that the FDR cutoffs and target-decoy analysis (see point 4.5 above) are not consistent with a high likelihood of chance matches.

4.7) *3) Despite the theoretical validation and given the relatively low number of peptides/spectra to investigate, have the authors considered doing a validation by a) synthetic peptides, b) predicted spectra or c) manual identification? To allow manual investigation, the authors could use tools such as xiSPEC to share annotated spectra of the novel peptides.*

This is a great suggestion – it would be a fantastic next step to validate the initial proteomic screen with additional experiments. However, for the reasons provided above, we are confident

that our analysis pipeline was stringent and resistant to false positives – at least to a commonly-accepted standard in the proteomics field. Therefore, given the relatively minor role of the mass spectrometry data in the overall story conveyed by this already-broad manuscript, we would argue that additional follow-up validation is beyond the scope of this work. To assuage any concern that our claims might extend beyond the experiments we performed, we modified the sentence concluding the paragraph describing the proteomics results (Results, lines 254-5). The word “demonstrate” was changed to “strongly suggest” so that the sentence now reads: “These findings *strongly suggest* that at least some of the predicted proteins are expressed on the surface of retinal cells in vivo.”

REVIEWERS' COMMENTS:

Reviewer #4 (Remarks to the Author):

The authors have addressed my concerns adequately by extending the manuscript and adding additional analysis. The data does support the finding of novel peptides.

I support the publication of the manuscript in principle.